# Mangrove removal exacerbates estuarine infilling through landscape-scale bio-morphodynamic feedbacks

Danghan Xie ●[1,2] ✉, Christian Schwarz ●[3,4], Maarten G. Kleinhans ●[1], Karin R. Bryan ●[5], Giovanni Coco[6], Stephen Hunt[7] & Barend van Maanen ●[8]

Changes in upstream land-use have significantly transformed downstream coastal ecosystems around the globe. Restoration of coastal ecosystems often focuses on local-scale processes, thereby overlooking landscape-scale interactions that can ultimately determine restoration outcomes. Here we use an idealized bio-morphodynamic model, based on estuaries in New Zealand, to investigate the effects of both increased sediment inputs caused by upstream deforestation following European settlement and mangrove removal on estuarine morphology. Our results show that coastal mangrove removal initiatives, guided by knowledge on local-scale bio-morphodynamic feedbacks, cannot mitigate estuarine mud-infilling and restore antecedent sandy ecosystems. Unexpectedly, removal of mangroves enhances estuary-scale sediment trapping due to altered sedimentation patterns. Only reductions in upstream sediment supply can limit estuarine muddification. Our study demonstrates that bio-morphodynamic feedbacks can have contrasting effects at local and estuary scales. Consequently, human interventions like vegetation removal can lead to counterintuitive responses in estuarine landscape behavior that impede restoration efforts, highlighting that more holistic management approaches are needed.

Coastal wetlands are crucially important but under pressure from a range of different drivers including changing sediment availability. The loss of coastal wetlands due to the shortage of riverine sediment supply (driven by human activities such as dam construction) in combination with sea-level rise has received significant attention in recent decades[1,2], while relatively less attention has been devoted to coastal wetlands with excessive sediment supply[3,4]. Over the last few centuries, land-use changes and coastal development have markedly increased sediment supply to the coast in many parts of the world[5–7], leading to substantial physical transformations of coastal landscapes[4] and ecosystems[3,8]. Such increases in sediment load and accelerated sedimentation in coastal areas, following large-scale catchment

deforestation, jeopardize pristine ecosystems and affect ecosystem services[9,10]. Paleorecords indicate that long-term fluvial sediment supply leads to the natural gradual infilling of estuaries, but human activities can accelerate this process causing more rapid coastal progradation and the expansion of coastal wetlands such as mangrove forests and salt marshes[3,11,12]. Wetlands, in turn, can stabilize fine sediment and are believed to accelerate the infilling of estuarine environments[13]. Although coastal restoration typically involves wetland re-establishment, restoration in rapidly infilling ecosystems often focuses on vegetation clearance[14–17]. For instance, the removal of rapidly expanding wetland vegetation aims to protect habitat diversity and is thought to promote the restoration of open estuary sand

[1]Department of Physical Geography, Utrecht University, Utrecht, the Netherlands. [2]Department of Earth and Environment, Boston University, Boston, USA. [3]Hydraulics and Geotechnics, Department of Civil Engineering, KU Leuven, Leuven, Belgium. [4]Department of Earth and Environmental Sciences, KU Leuven, Leuven, Belgium. [5]School of Science, University of Waikato, Hamilton, New Zealand. [6]School of Environment, University of Auckland, Auckland, New Zealand. [7]Waikato Regional Council, Hamilton, New Zealand. [8]Department of Geography, University of Exeter, Exeter, UK. ✉e-mail: danghan@bu.edu

dominated systems which are of high socio-economic value[15]. The question of how to manage wetland vegetation in accreting coastal environments is highly relevant especially given the increasing level of human-induced disturbances in such systems, where it remains unclear whether local interventions can reverse impacted ecosystem states to previous more diverse and ecosystem service-rich conditions[18–20].

Conditions of estuarine systems in New Zealand can be used as a test case to explore how poor understanding of vegetation effects at the estuary scale can compromise restoration efforts. Following European settlement, most upstream regions of the North Island of New Zealand experienced substantial conversions of forestland to agriculture or pastures, resulting in rapid and widespread soil erosion in the hinterland[21]. These land-use changes caused at least an order-of-magnitude increase of suspended sediment yields to the coast compared to pre-European times[22,23]. Sedimentation in estuaries therefore magnified from relatively low rates (0.1–1 mm/yr prior to European

settlement) to continually increasing rates (with maximum contemporary values exceeding 100 mm/yr) (Fig. 1f)[21,24,25], creating widespread accumulation of intertidal mud and causing rapid mangrove expansion (Fig. 1e). Such excessive sediment deposition and mangrove expansion not only impact navigation, limit recreational activities and the amenity value of these areas[15], but also transform coastal habitats at the expense of highly valuable low-turbidity ecosystems, such as those dominated by seagrass and filter-feeding shellfish[10]. Given these perceived negative impacts on the estuarine environment, a pro-removal attitude has developed amongst some communities and both legal and illegal mangrove clearance has occurred in recent years to restore pre-disturbed conditions[15,19,20]. At the same time, the unique characteristics of mangrove forests are increasingly being recognized and the public view on mangrove expansion and removal thus remains polarized[26]. The ongoing debate is fed by the uncertainty on restoration success, which is directly related to a lack of understanding on the effects of vegetation removal on estuary-scale morphological changes.

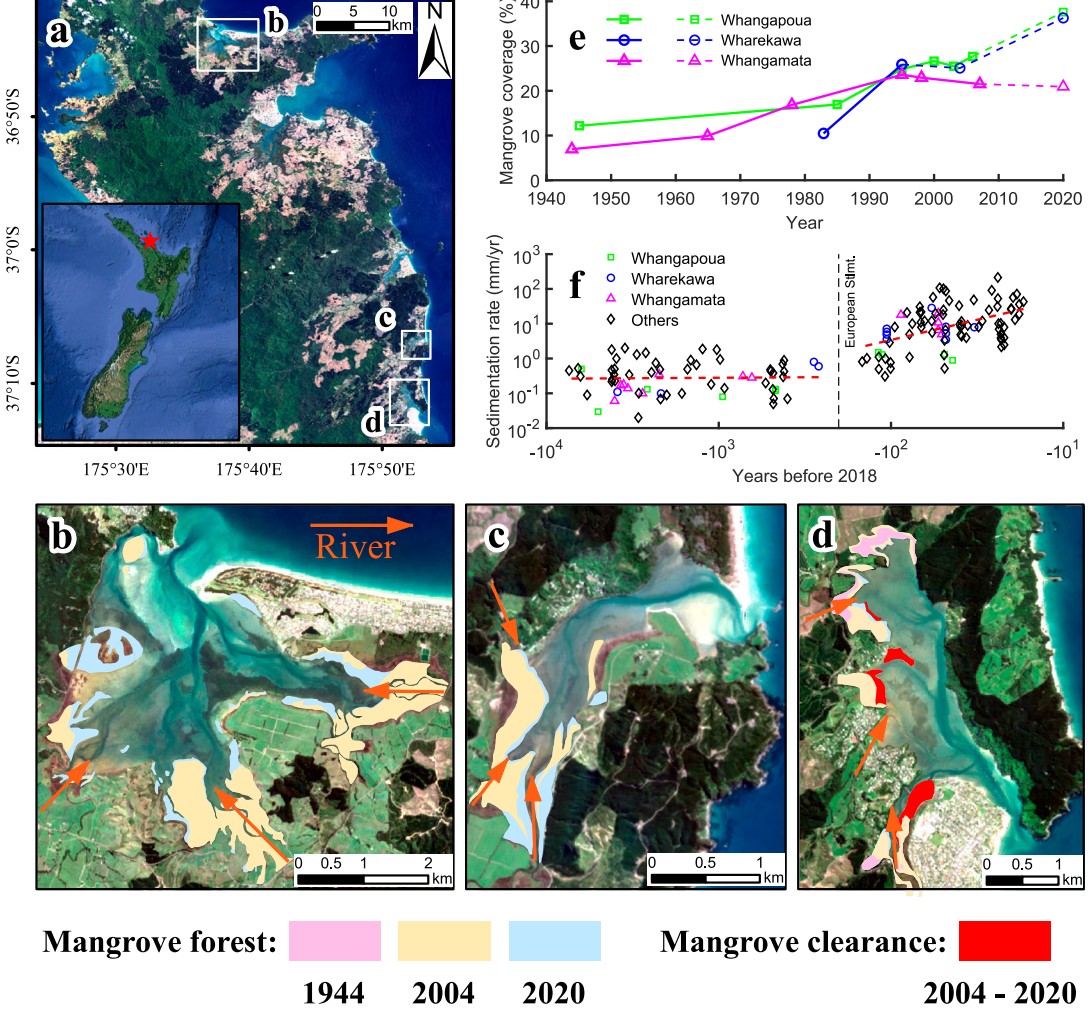

**Fig. 1 | Temporal variation in mangrove distribution and sediment accumulation rates at three representative estuaries on the North Island of New Zealand.** (**a**) Location map for estuaries shown in (**b**–**d**). (**b**) Whangapoua estuary; (**c**) Wharekawa estuary; (**d**) Whangamatā estuary. (**e**) Observed changes in mangrove coverage at these three estuaries. (**f**) Historical sediment accumulation rates. The orange arrows in (**b**–**d**) indicate riverine input. Mangrove coverage refers to the percentage of mangrove presence relative to the estuarine area. Mangrove coverage data in solid lines in (**e**) is derived from Jones[114] and data in dashed lines are estimated from recent Landsat data. The reduction in mangrove cover in the Whangamatā estuary is due to mangrove clearance, also see (**d**). Mangrove distributions in year 2004 and 2020 are based on Landsat data from Giri et al.[115] and datasets from Land Information New Zealand (https://data.linz.govt.nz/). Contains data sourced from the LINZ Data Service licensed for reuse under CC BY 4.0. Mangrove distribution in 1944 in (**d**) is based on historical archive data published in Lundquist et al.[15]. Mangrove clearance area is taken from datasets in Bulmer et al.[55]. Historical sediment accumulation rates (**f**) are based on datasets compiled by Hunt[24].

In recent decades, bio-morphodynamic feedbacks have been increasingly shown to shape coastal landscape evolution as vegetation interacts with water flow and sediment transport[27–30]. Vegetation locally enhances hydraulic resistance, which then reduces flow velocity and increases sediment deposition[31–33]. In addition, vegetation stabilizes mud deposits and thus optimizes conditions for seedling colonization[34–36]. Given such interactions, mangrove vegetation should accelerate mud accumulation and promote mangrove growth[37]. Indeed, to halt and revert muddification of New Zealand estuaries, knowledge of such local-scale bio-morphodynamic feedbacks has been applied to restore coastal ecosystems and create incentives for mangrove removal within rapidly infilling systems[13,15,38]. However, estuaries do not only host homogenous vegetated areas, but instead consist of intricate networks of tidal channels, tidal flats and vegetated platforms[30]. These interconnected landforms in turn have significant effects on bio-morphodynamic development at the landscape scale, i.e. the estuary scale, causing complex flow and sedimentation patterns, impacts of which have not been explored in detail. Furthermore, rather than significantly increasing sedimentation rates, it has been suggested that mangroves are opportunistic and colonize areas that have already reached a suitable intertidal elevation through historic sedimentation[39,40]. Removing mangroves may therefore not measurably reduce sediment trapping and sedimentation, but rather lead to a redistribution of water flow and sediment with unknown effects on estuary-scale development. Due to limitations in studying various temporal and spatial scales in the field post mangrove removal, insights thus far remain inconclusive[15,41]. Results of bio-morphodynamic model predictions can fill this gap, by accounting for small-scale interactions between vegetation, hydrodynamic forces and sediment transport, and the emergent effects of these bio-morphodynamic feedbacks on estuary-scale development.

In this work, we investigate the effects of changing sediment supply on estuarine bio-geomorphic landscape development with a focus on whether local measures, like mangrove removal, can reduce mud infilling and potentially restore ecosystems in estuaries with a history of anthropogenically increased fine sediment input. We assess whether established knowledge of local-scale bio-morphodynamic feedbacks can be extrapolated to the estuary scale to deliver anticipated restoration outcomes. The idealized model setup used here represents a back-barrier estuary with multiple river inputs as often observed in estuarine systems globally, including the North Island of New Zealand (Fig. 1). Model development and simulation include several steps. We first establish a sandy estuarine basin with an average platform elevation of −1.5 m relative to mean sea level. This is followed by a period of low mud supply from three rivers discharging into the basin representing pre-disturbed conditions. Subsequently, the muddy sediment load of the rivers is increased representing the impact of catchment deforestation following the arrival of European settlers, transitioning the landscape into a highly disturbed state. Then, to forecast the impact of different management scenarios focussing on the strategy of either upstream land-use or in-situ estuarine interventions, we adjusted the mud supply and simulated events of mangrove clearance. We find that the removal of mangroves does not limit estuarine sediment trapping but in fact increases mud infilling due to the scale-dependency of bio-morphodynamic feedbacks, suggesting that a catchment-focus to estuary management is necessary.

## Results

### Impacts of increased sediment supply

An increase in fluvial sediment supply led to significant differences in morphological development and mangrove distribution in the estuary. At the pre-disturbance stage (year 200–400), mangroves first colonized levees (i.e. elevated areas along channels) close to the river mouths where sediment from catchments was deposited (Fig. 2b). The morphological evolution was characterized by coastal progradation, where fine sediment continued to deposit at the seaward edge, creating intertidal areas and channels carving through the estuarine basin (Fig. 2b). Over these 200 years, mangroves slowly expanded seaward as inundation regimes became favorable, reaching a coverage of 8.78% over the basin area at the end of the pre-disturbance period. During the disturbance period, with a high mud supply (year 400-500), morphological evolution was dominated by vertical mud accumulation. Estuarine infilling was accelerated with further sedimentation on the intertidal areas, leading to a seaward expansion of mangrove forests along channels (Fig. 2c). The proportion of mangroves covering the estuarine basin nearly tripled within 100 years under high sediment loading (21.59%) (Fig. 2c).

### Estuary-scale changes under management scenarios

To understand the impact of management actions on the estuary, we investigated the scenario of continued high mud supply in combination with different mangrove removal strategies from year 500, including no removal (Fig. 2d) and 25%, 50% and 100% mangrove coverage removal (Fig. 2e, h, k). In addition, we simulated cases in which mud supply was reduced to pre-disturbance (Fig. 2g) and intermediate levels (Fig. 2j) to explore the effects in case that more sustainable catchment use would be reinstated. Such mud supply reductions were also implemented earlier (i.e. from year 430 and 450) to explore the effects of disturbance duration (Fig. 2f, i). A control run with continued low mud supply throughout the simulation was used for comparison (not shown).

We found that changes in catchment sediment yield had a much stronger control on key characteristics of estuarine landscapes than the removal of mangrove vegetation (Figs. 3 and S1). An increase in mud supply from pre-disturbed conditions accelerated the infilling of accommodation space (Fig. 3a) and the creation of muddy regions within the estuary (Fig. 3b), concomitantly resulting in faster mangrove expansion (Fig. 3c). Reducing the mud supply back to lower levels helped decelerate the rate of accommodation space infilling and development of mud areas, thus slowing mangrove expansion (Fig. 3). The magnitude and timing of mud reductions played a key role in determining the evolving state of the ecosystem. That is, accommodation space, the extent of the muddy region and mangrove coverage remained comparable with the situation under continued low mud supply (yellow line in Fig. 3) when the mud reduction occurred after only a short disturbance period (e.g. yellow-circle line). A delayed (e.g. yellow-triangle line) or a smaller (e.g. dashed yellow line) reduction in mud supply allowed the bio-geomorphic characteristics of the estuary to deviate further from the undisturbed system.

A current management practice in New Zealand estuaries involved localized mangrove clearance but model simulations indicated that this intervention did not limit ongoing muddification. Mangrove removal in fact enhanced estuarine infilling and resulted in a larger portion of the estuary consisting of muddy substrate (Fig. 3a, b). These effects were exacerbated with increasing levels of mangrove removal. Thus, when mangroves were completely removed, the accommodation space became smaller, with an increased muddy area compared to scenarios of partial or no mangrove removal (see inserts in Fig. 3a, b and also Fig. S1a, b).

### Changing sedimentation patterns

The removal of mangrove vegetation caused changes in sediment dynamics in both channelized and unchannelized areas of the estuary. Within the channels, we found that mangrove removal resulted in a larger mud thickness (green violin in Fig. 4c) with less sediment erosion (green violin in Fig. 4e), leading to a relatively higher bed elevation compared with other management scenarios (green violin in Fig. 4a). In unchannelized areas, sedimentation rates varied with distance from channels, and the spatial trends were reversed when mangroves were removed. More specifically, when mangroves were present,

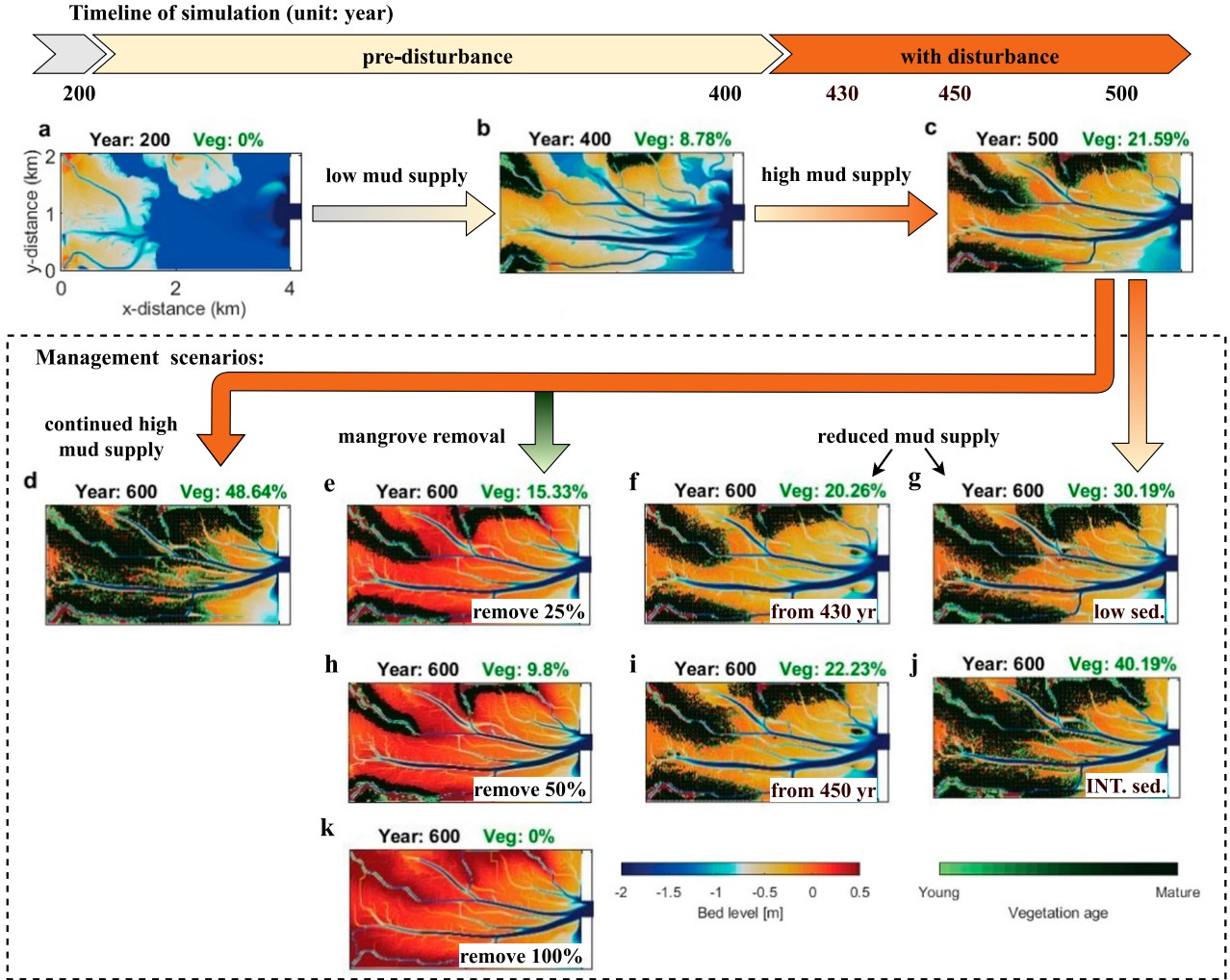

**Fig. 2 | Mangrove distribution and morphological development phases.** The vegetation cover as a fraction of estuarine basin area is indicated as a green number above each panel. Morphology after 200-year spin up (**a**), pre-disturbance with low mud supply and limited mangrove colonization (**b**), disturbance with high mud supply and rapid mangrove expansion (**c**). Management scenarios include: (**d**) continued high mud supply; (**e**, **h**, **k**) removal of 25%, 50% and 100% of mangroves; reduced mud supply from year 430 (**f**) and year 450 (**i**); reduced mud supply to its pre-disturbed lower level (**g**) and an intermediate level (**j**) from year 500. yr = year; low sed. = low sediment supply; INT. sed. = intermediate sediment supply.

sedimentation rates were relatively high in the proximity of channels and diminished further away from channels (purple violins in Fig. 4f). However, after mangrove removal, sedimentation rates were lower near channels and became larger in more distant areas of the estuary (green violins in Fig. 4f). Such changes in sedimentation rates driven by mangrove removal led to a higher mud thickness and bed elevation of the flats further away from channels (purple and green violins in Fig. 4b, d). As a consequence, the overall area with muddy substrates in the estuary might not reduce, but rather increase through mangrove removal (see also Fig. 3b). Under reduced mud supply (blue violins in Fig. 4), model results showed that both channelized and unchannelized areas would retain more similar bed elevations, mud thicknesses and sedimentation rates to those observed at the pre-disturbance stage.

**Estuarine landscape trajectories**

To relate basin response directly to catchment management and disturbance, we evaluated the relative basin area above mean sea level (aMSL) against the non-dimensional catchment sediment yield (Fig. 5). Large basin areas aMSL resembled infilled estuaries with extensive potential mangrove habitats, while small areas aMSL corresponded to unfilled systems dominated by unvegetated flats and subtidal areas.

Model results followed previously described estuarine infilling patterns driven by fluvial sediment yields found in New Zealand[21]. We found that estuarine landscapes deviated strongly from their pre-disturbed state when an increased mud supply was maintained, causing relative basin area aMSL to exceed 0.5 (Fig. 5*). This implied that more than half of the estuarine area was suitable for mangrove growth. Relative basin area aMSL remained below 0.2 for the simulated estuary with continued low mud supply (□). Removal of mangroves slightly enhanced the formation of upper intertidal area and thus unexpectedly facilitated the creation of new mangrove habitat (◇). Model simulations showed that estuarine infilling and intertidal area development could be limited through reductions in sediment supply, but the magnitude and especially timing of such reductions were critical as the dependency of basin areas aMSL on sediment supply was not linear. For example, when sediment yield was reduced back to the pre-disturbed condition after 100 years, basin areas aMSL increased more rapidly during subsequent lower sediment yield (from 0.27 to 0.36 between 500 and 600 years; Δ) than when sediment yield was restored after 50 years (from 0.17 to 0.26 between 450 and 600 years; O). This highlighted that estuarine landscapes were not only modified throughout the disturbance period, but that also longer-term effects on landscape trajectories were controlled by disturbance duration.

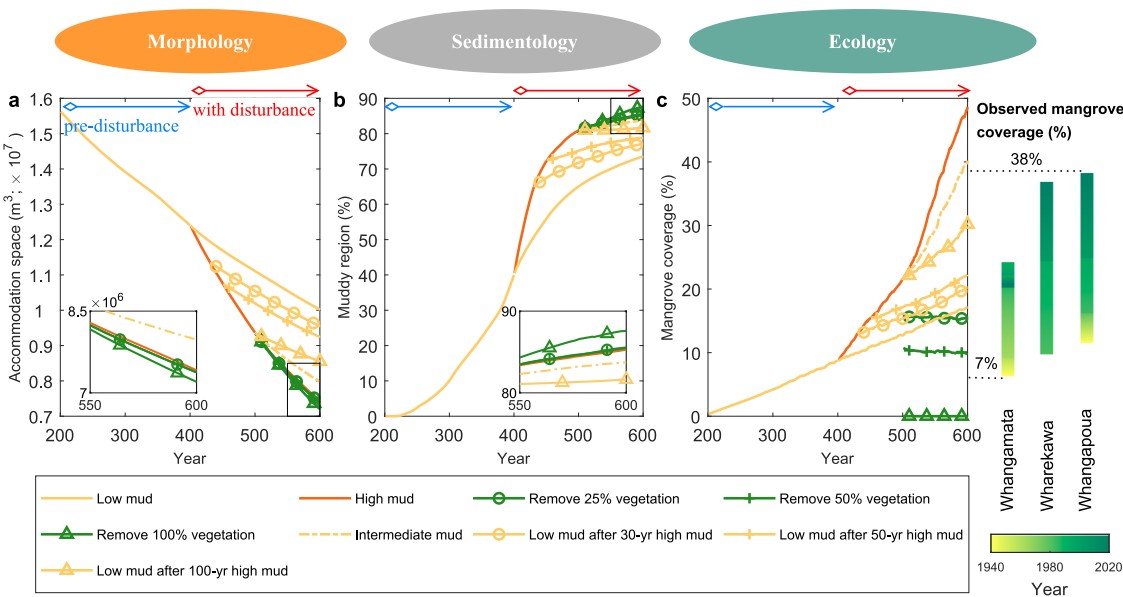

**Fig. 3 | Temporal changes of key morphological, sedimentological and ecological characteristics of the estuarine environment.** Different scenarios of mud supply and mangrove removal are presented. (**a**) Accommodation space calculated as the total basin volume below high tide that could be filled with sediment, (**b**) muddy region defined as the relative surface area for which the mud fraction in the top 1-m profile is larger than 30%[116] and (**c**) mangrove coverage calculated as a fraction of the basin area colonized by mangrove forests.

## Discussion

Our results suggest that variations in upstream sediment supply are the main driver of estuarine muddification and mangrove coverage change. Moreover, the outcome of ecosystem restoration measures is determined by the scale-dependency of bio-morphodynamic feedbacks (Fig. 6). Mangrove vegetation is known to locally reduce tidal currents and therefore facilitates mud deposition and bed accretion[42,43]. This in turn enhances mangrove growth, forming a reinforcing bio-morphodynamic feedback loop at the local scale (Fig. 6). The effects of this feedback loop can also be observed in our simulations through the development of profound levees as mangroves constitute effective sediment traps near tidal channels. Based on such understanding of local-scale bio-morphodynamic processes, i.e. mangroves enhance sediment trapping, mangrove removal is expected to reduce overall mud trapping and potentially restore estuarine sand flats/beaches present prior to mangrove colonization.

Our numerical experiments give reason to question this paradigm, and show that at the estuary scale, mangrove removal in fact reinforces estuarine mud infilling and intertidal habitat creation through reconfiguration of landscape flow and sedimentation/erosion patterns (reinforcing anthro-bio-morphodynamic feedback loops at the estuary scale in Fig. 6). Mangrove vegetation causes water flow to be conveyed predominantly through channels[44,45]. Mangrove removal thus leads to relatively reduced current strength within channels (Fig. S2a), thus, enhancing sedimentation and channel infilling (Figs. 4a, S2e, S4). At the same time, mangrove removal increases currents and sediment delivery to the tidal flats (Fig. S2b, d), facilitating enhanced sediment accretion compared to intact mangrove forests, especially in more distant regions within the estuary located further from channels (Figs. 4b, S2f, S4). Therefore, complete mangrove removal resulted in enhanced estuarine infilling, less accommodation space and further muddification exemplified by a larger muddy region within the basin (Figs. 3, S1). Also, when comparing additional model runs with and without any vegetation throughout the simulation (Fig. S5), we find the presence of mangroves limits estuarine infilling due to concentrating sedimentation on the levees adjacent to channels, creating a balancing bio-morphodynamic feedback loop at the estuary scale (Fig. 6). Human interventions, through mangrove removal, then convert this into a reinforcing feedback loop that accelerates estuarine infilling.

Since mangrove removal reinforces estuarine infilling, the existing mangrove removal strategy may not be appropriate for restoring ecosystems but rather aggravate the muddification of estuaries. Our model results advocate for a focus shift in coastal management of fast infilling systems caused by high sediment supply from upstream (Fig. 6). Our simulations show that a reduction in upstream mud supply is able to reduce mud accumulation and mangrove expansion rates. Moreover, the estuarine configuration resulting from infilling is not only linked to the level of mud reduction but also the timing as this determines disturbance duration (Figs. 3, 5). More specifically, adjusting the timing of mud reductions can lead to vastly different percentages in intertidal area and thus mangrove coverage (Fig. 5 O vs. Δ) and a reduction "early" in the development can have major implications on slowing down infilling. This implies that we should urgently transition away from mangrove removal as management approach. Reinstating more sustainable upstream land-use should restore catchment forest cover and thus reduce soil erosion and sediment delivery to estuarine systems, creating a balancing feedback loop at the source-to-sink scale (Fig. 6). Future coastal management strategies should thus not only focus on actions in the estuary itself but instead adopt a whole-system view (i.e. source-to-sink) that incorporates interconnections between human activity, morphological changes and biological impacts. Model-derived estimates of temporal scales of landscape development in response to changed external forcings could give indications of the strength of this system's feedback loop and guide management decisions.

Our idealized model simulations capture general infilling trends and can be used to explore the impacts of human disturbances and different management strategies. The effect of anthropogenically enhanced sediment supply on estuarine bio-geomorphic development is likely to be still conservatively estimated in our simulations. Historical data suggest that mangrove coverage of the three New Zealand estuaries shown in Fig. 1 increased at a much faster rate than in the model (Fig. 3c). As an example, within the Whangapoua estuary, mangrove coverage increased from 10% to ~40% within 80 years, while it took nearly 140 years for our models to reach the same coverage

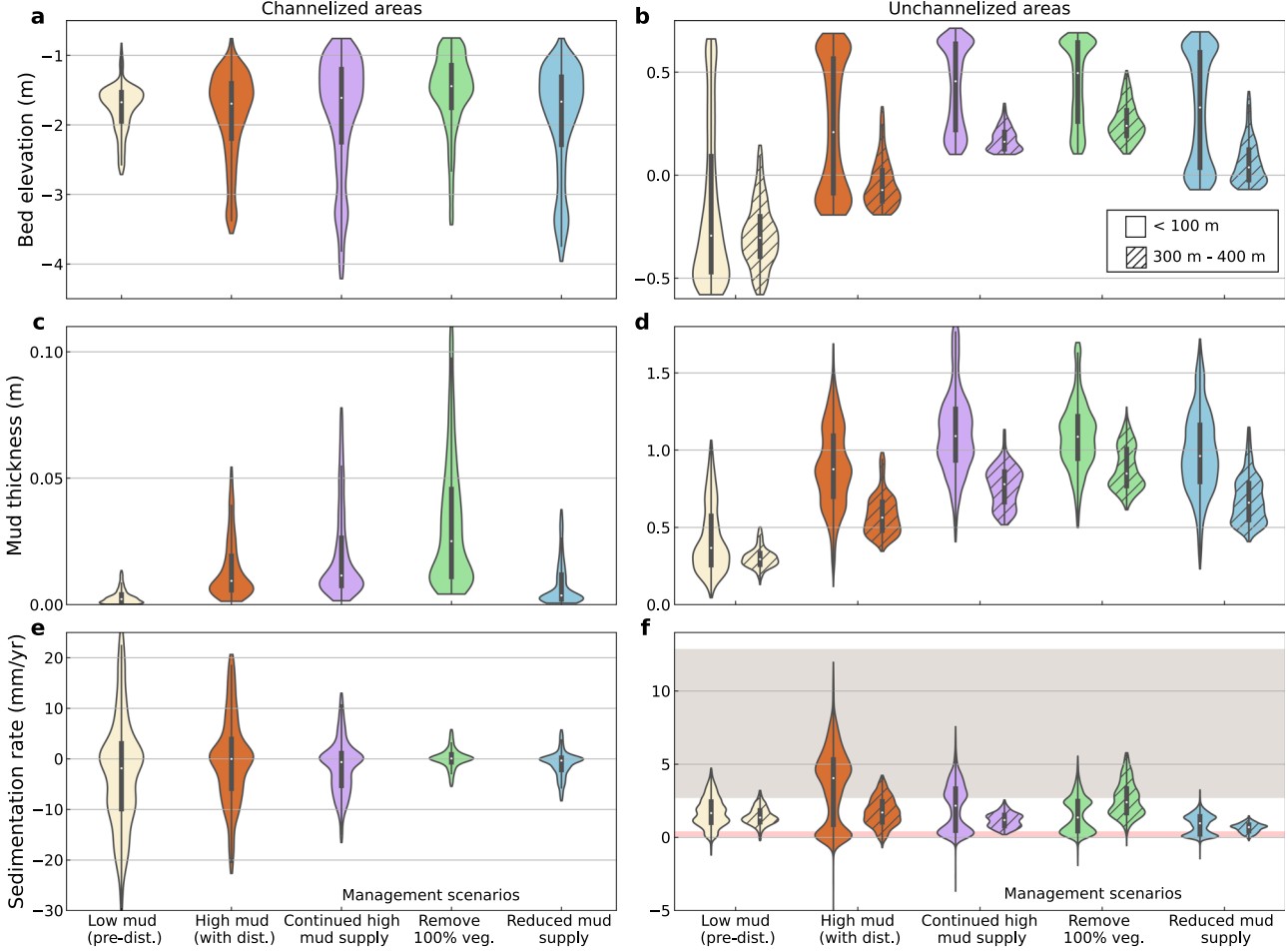

**Fig. 4 | Comparison of the distribution of bed surface properties expressed as mean bed elevation, mud thickness and sedimentation rate for five representative scenarios in Fig. 2.** The comparisons are conducted for both channelized (**a**, **c**, **e**) and unchannelized areas (**b**, **d**, **f**), the latter of which have been further categorized into two classes based on the distance to the nearest channels (i.e. platform areas close (<100 m) and further away from channels (300-400 m)). Low mud scenario indicates the stage before European settlement accompanied by a limited amount of mud supply. High mud scenario represents the system disturbed by a large mud supply after European arrivals. Three possible management strategies following high mud supply scenario, with continued high mud supply, remove mangrove vegetation (100%), or reduce mud supply, are also listed in the plots. Yellow, orange, purple, green and blue colors are used to represent the scenarios in low mud (corresponding to Fig. 2b), high mud (corresponding to Fig. 2c), continued high mud (corresponding to Fig. 2d), remove vegetation (corresponding to Fig. 2k), and reduced mud supply (corresponding to Fig. 2g). Violin thickness corresponds to probability density. Endpoints of violin depict minimum and maximum values. Box plot inside each violin covers the first to third quartiles, with a diamond representing the median value. Pink and gray shadings in f indicate the observed sedimentation rate range (99% confidence interval) at Whangapoua, Wharekawa and Whangamatā estuaries before and after European settlement, respectively (Fig. 1f).

changes, implying that the real estuarine infilling rate was greater than predicted with our model settings. Maximum measured accumulation rates post European settlements (Fig. 1f) also exceed typical accumulation rates observed in our high mud supply scenarios (Fig. 4f). At the same time, our model does not account for wave action or high river discharge events while these processes may influence sediment resuspension and estuarine infilling trends[46–48]. As our simulations focus on sheltered estuaries with relatively small upstream catchments (Fig. 1), limited wave action and peak discharges can be reasonably assumed. For larger and less sheltered estuarine systems where wave action may be amplified, or those estuaries that are subject to more extreme discharge events, sediment resuspension after mangrove removal may be more likely. Still, an analysis of mangrove removal sites on the east coast of New Zealand suggests that more exposed sites only account for a limited proportion of total mangrove removal areas[13,15], suggesting that mangrove clearance is not a viable approach. In addition, sediment compaction and the on-going presence of mangrove roots after mangrove removal, both of which increase the erosion threshold, are also neglected in our models but have been found to further limit sediment resuspension[15,49,50], and thus hinder the potential for mud export.

In this study, the infilling of accommodation space is fully driven by mineral sedimentation, while organic accretion driven by root production is not included. We acknowledge that belowground root growth can be an important or even dominant process controlling surface elevation change[51]. According to field observations from multiple mangrove sites, belowground root induced surface accretion tends to show a linear relationship with root production (Fig. S13). However, local data shows that belowground root production in New Zealand mangrove forests is smaller ($50\,g/m^2/yr$) than in most other tropical mangrove sites, contributing less than 1% sedimentation volume[52]. Such a limited root production is expected to result in a negligible accretion rate (less than 0.5 mm/yr, Fig. S13). Mangrove dieback has also been found to drive root collapse, which in turn lowers surface elevation and increases accommodation space[53,54]. However, studies in New Zealand indicate that limited changes occurred in the surface elevation after mangrove removal[13,55,56]. This is probably because of slow decomposition rates in soil organic

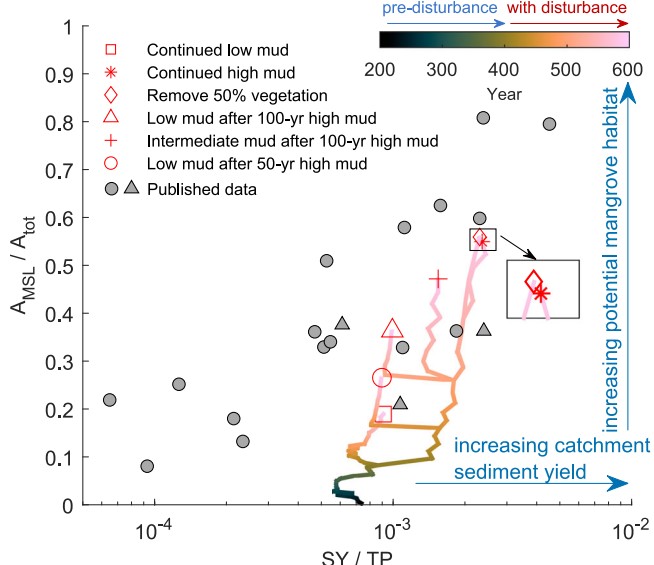

**Fig. 5 | Changes of relative basin area above mean sea level ($A_{MSL}/A_{tot}$) as a function of the non-dimensional catchment sediment yield (SY/TP).** The simulated annual catchment sediment yield (SY) is the total sediment load from the three rivers. The tidal prism (TP) was calculated as the water discharged through the inlet during one tidal cycle. $A_{MSL}/A_{tot}$ represents the potential mangrove habitat and is calculated as the ratio between basin area above mean sea level ($A_{MSL}$) and total basin area ($A_{tot}$). Six model simulations are shown to indicate different estuarine landscape trajectories. The gray triangles represent the three estuarine systems shown in Fig. 1 using data from Hicks et al. [22] and Hume and Herdendorf[101]. The gray circles represent other New Zealand systems based on Swales et al. [108]. More details can be seen in Table S2 in the Supplementary Information 1.pdf.

matter[15,57] and limited belowground root biomass as well as root production compared to other tropical mangrove sites[52,58]. Apart from belowground processes, sea-level rise can also affect changes in accommodation space. Here we did not consider the impacts of sea-level rise on estuarine infilling given the low sea-level rise rate (around 1.4 mm/yr), compared to the high sedimentation rate (10–100 mm/yr)[59]. However, projected accelerations in sea-level rise may create additional accommodation space[60,61], thus slowing down the estuarine infilling process. Sedimentation rates along vegetated tidal flats have been found to be non-linearly related to sea-level rise rates[42], such that future estuarine infilling is likely to be a complex process that needs to be further explored[62].

The importance of mangrove removal on sedimentation patterns and estuarine infilling as found by our models is underpinned by recent studies that reveal vegetation effects for other types of coastal systems. Specifically, for deltaic salt marshes, contrasting effects on sedimentation patterns have been revealed, whereby vegetation can enhance sedimentation but also divert flow away from dense vegetation so that sediment deposition is reduced[63,64]. The ability of vegetation to confine flow and sediment within channels has also been found to reduce total deltaic sediment retention[65]. For a freshwater tidal marsh, vegetation removal experiments have shown that vegetation clearance reduced channel flow velocities while increasing flow velocities on the platform[66]. This causes a spatial redistribution of sediment with the potential of enhanced sedimentation in the inner marsh[67] which is in agreement with our model findings.

Furthermore, the presence of vegetation has previously been found to promote levee growth in fluvial-tidal environments[68]. Levee development is of critical importance as levees not only store sediment locally directly adjacent to channels, but also because they influence channel hydro-sedimentary processes and the delivery of sediment to the vegetated platform[68–70]. Our research reveals that

vegetation effects on levee formation and associated channel flow depend on along-channel location and whether the channels are dominated by tides or river flow (Figs. S9–11). For river-dominated channels, vegetation strengthens seaward directed flows and can even suppress flow reversals during the flood tide (Fig. S10). As such, river-dominated channels flanked by vegetation become more effective conduits for water and sediment transport, with implications for estuarine sediment budgets. For channels driven by river flow, levee height generally decreases with increasing distance from the sediment source. Mangroves are found to promote levee formation but only at downstream locations. In contrast, for tide-dominated channels, vegetation mainly enhances flood currents and contributes to levee elevation but here only at the more upstream locations (Fig. S11). Clearly, vegetation effects on sedimentation patterns and changes following vegetation removal are not simply determined by the lateral distance from a channel, but can spatially vary and depend on the dominant drivers of individual channel distributaries. Mutual interactions between vegetation, vegetation-influenced landforms and sedimentation patterns are extremely complex and deserve further attention given the significant implications for management and ecosystem resilience.

While global mangrove loss has received much attention in recent years[71,72], mangrove expansion has been observed in many places around the world. This is not only because of increasing sediment supply following land-use change (e.g. New Zealand, Australia, Hawaii, Hong Kong)[13,14,17,19], but also through climate change driven ecotone shifts (e.g. Australia and Florida)[73] and man-made introduction of particular mangrove species (e.g. Taiwan)[16]. Similar reports exist for salt marsh expansion in historic Europe and North America, and recent China[3,18,74]. Public concerns about such changes in coastal ecosystems and highly contrasting views on vegetation removal increases the demand on sustainable and well-founded management approaches. In stark contrast, in vegetation-sparse coastal systems where ongoing erosion is typically the main concern, vegetation re-establishment is carried out where Building with Nature projects were able to show promising results, raising the question whether these approaches can be upscaled[75]. However, bio-morphodynamic effects on larger spatial and temporal scales are still largely unknown. Our study highlights that, in complex and highly coupled human-biogeomorphic systems, any interventions related to either vegetation removal or re-establishment requires a multi-scale assessment of bio-morphodynamic feedbacks ranging from local effects to emerging effects at the coastal landscape scale, to ensure that restoration efforts and human interventions more broadly are delivering anticipated outcomes.

## Methods
We extend a previously developed one-dimensional bio-morphodynamic model[42,76] to two-dimensions to capture spatial mangrove behaviors and sediment dynamics in an estuary, specifically a back-barrier fluvial-tidal basin. The two-dimensional bio-morphodynamic model is composed of a hydro-morphodynamic model (in Delft3D, version 4.01.00) and a dynamic vegetation model (in Matlab, version R2017a), which are connected seasonally.

### Hydro-morphodynamic processes
Delft3D, a morphodynamic modeling package, simulates the water level and flow velocity by solving the depth-averaged shallow water equations[77,78]. The presence of vegetation is incorporated by including additional hydraulic resistance through calculation of the bed roughness ($C_r$) and the additional resistance term ($-\frac{\lambda}{2}u^2, -\frac{\lambda}{2}v^2$). Both $\lambda$ and $C_r$ are derived from vegetation characteristics (diameter, height, density) and will be introduced in the next section (Eqs. 1–2).

Following previous observations in New Zealand estuaries, the model considers both cohesive (mud) and non-cohesive (sand)

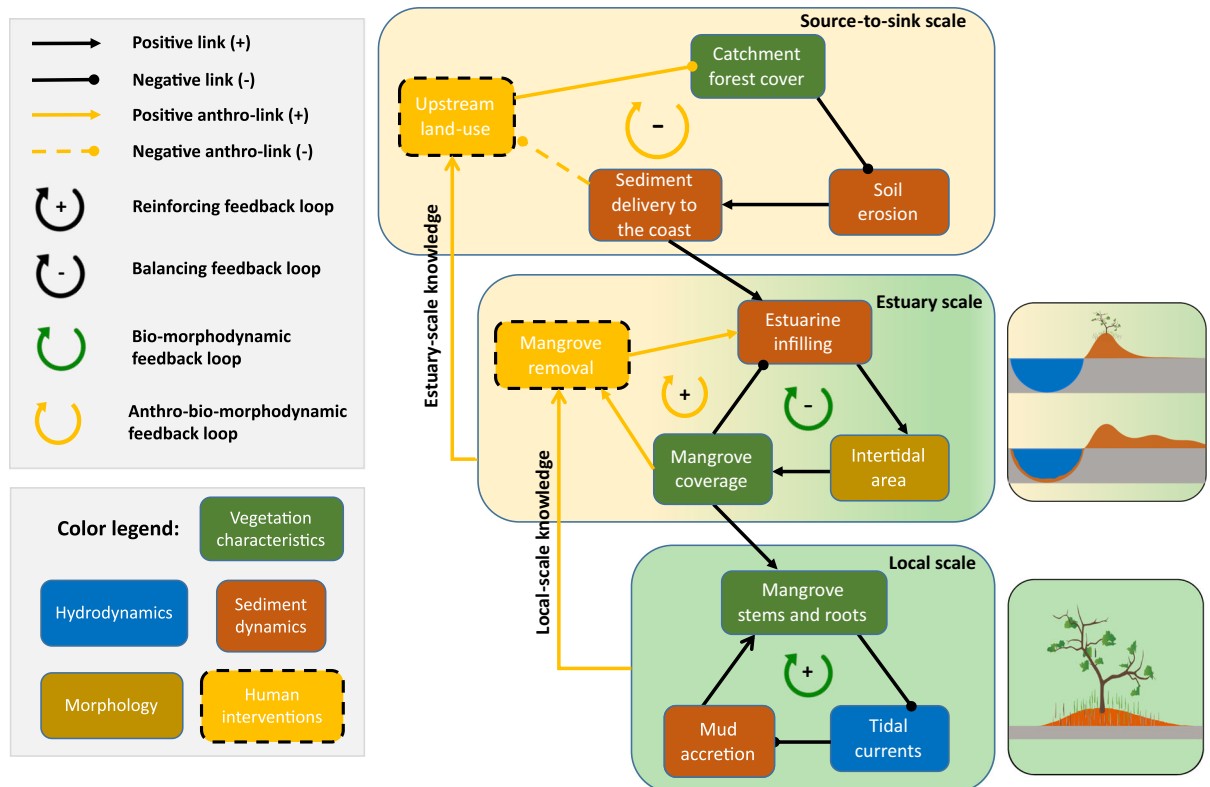

**Fig. 6 | Conceptual diagram outlining distinct bio-morphodynamic and anthro-bio-morphodynamic feedbacks at the local, estuary and source-to-sink scale.** Reinforcing and balancing feedback loops[117] are indicated. Here, 'upstream land-use' refers to pastoral farming and agriculture which result in large-scale catchment deforestation. 'Positive link' and 'negative link' refer to positive and negative correlations, respectively. Dashed lines surrounding the rectangles for 'Human interventions' represent either a one-time intervention or more continuous interventions that trigger the feedback loop. The dashed line indicating the link from 'sediment delivery to the coast' to 'upstream land-use' suggests a more sustainable and effective management approach that addresses source-to-sink linkages. The mangrove icon is sourced from the Integration and Application Network (ian.umces.edu/media-library) under the CC BY-SA 4.0 license.

sediments[38,79]. For muddy sediment, the deposition and erosion fluxes are computed through the Partheniades–Krone formulation[80] with parameters setting consistent with recent field research[81]. For sandy sediment, the Van Rijn transport predictor is used[82,83] to calculate the suspended-load transport and bed-load transport separately. The transport of suspended sediment is calculated according to the advection-diffusion equation. The transverse bed slope effect on bed-load transport is parameterized with Koch and Flokstra formulations[84], after Baar et al. [85]. At every hydrodynamic time step, the changes of bed level are calculated based on the sediment mass balance considering deposition/erosion sediment fluxes and the bed-load transport in $x$ and $y$ directions. A morphological acceleration factor (here set to 90) is applied to enable long-term simulations based on a sensitivity analysis[28,86].

**Dynamic vegetation processes**

To account for the effects of vegetation on hydrodynamics, we quantify the vegetation-induced flow resistance through the Baptist predictor[87], which was implemented to allow for multiple fractions of different vegetation types in one numerical grid cell[88]. Based on the relative relations between the height of vegetation objects (such as stems or roots) $h_v$ (m) and local water depth $h$ (m), the bed roughness $C_r$ (m$^{1/2}$/s) is calculated as follows:

$$C_r = \begin{cases} C_b + \frac{\sqrt{g}}{\kappa}\ln\left(\frac{h}{h_v}\right)\sqrt{1 + \frac{C_D n h_v C_b^2}{2g}}, & if\ h \geq h_v \\ C_b, & if\ h < h_v \end{cases} \quad (1)$$

where $C_b$ is the Chézy coefficient for the unvegetated bed, set to 65 (m$^{1/2}$/s); $\kappa = 0.41$ is the Von Kármán constant; $C_D$ is the drag coefficient (dimensionless); $n$ is the vegetation density (m/m$^2$) calculated as $n = mD$ where $m$ is the number of vegetation objects per unit area (1/m$^2$) and $D$ is the diameter of this object (m).

Vegetation causes a higher hydraulic resistance which could then lead to a higher bed shear stress and larger sediment transport rates during morphological calculations. To correct this, Delft3D includes a term $(-\frac{\lambda}{2}u^2, -\frac{\lambda}{2}v^2)$ in the momentum equations, where $\lambda$ is calculated as:

$$\lambda = \begin{cases} C_D n \frac{h_v}{h}\frac{C_b^2}{C_r^2}, & if\ h \geq h_v \\ C_D n, & if\ h < h_v \end{cases} \quad (2)$$

The dynamic mangrove model includes colonization, growth and mortality based on our previous research[42,76]. Mangrove colonization occurs at the first ecological season when both inundation regime and current strength are appropriate for seedling settlement. As mangroves mainly occupy areas between mean water level and mean high water[89,90] and seedling establishment is hindered under larger bed shear stresses induced by currents/waves[91], we assign an initial vegetation density to the cells with relative hydroperiod ranging between 0 and 0.5, and bed shear stress below 0.2 N/m$^2$. The initial seedling density is set to 3000 individuals/ ha following van Maanen et al.[92]. Infilling of accommodation space due to sedimentation can suppress the growth of mangroves and result in a lower vegetation density if the upper limit of mangrove elevation is being reached[42]. The ecological

time to describe mangrove dynamics is set equal to the morphological time, such that within one morphological year the vegetation is updated four times (i.e. seasonally).

After initial settling, mangroves grow each ecological season which is evaluated through an increase in stem diameter ($D$; cm) and height ($H$; cm)[92–94]:

$$\frac{\mathrm{d}D}{\mathrm{d}t} = \frac{GD\left(1 - \frac{DH}{D_{\max}H_{\max}}\right)}{\left(274 + 3b_2D - 4b_3D^2\right)} \cdot f \cdot C \tag{3}$$

$$H = 137 + b_2D - b_3D^2 \tag{4}$$

where $t$ is the time (season), $D_{\max}$ and $H_{\max}$ are the maximum stem diameter and tree height, respectively. $G$, $b_2$ and $b_3$ are growth parameters (Table S1). Tree growth may be reduced by sub-optimal inundation conditions and because of limitations in available resources. This is incorporated through a fitness function ($f$) and the competition stress factor ($C$). Both $f$ and $C$ range between 0 (no growth) and 1 (optimal growth)[92,95]. $f$ is dependent on hydroperiod and $C$ is dependent on mangrove biomass[42,76].

At the end of every ecological year, the mortality process is initialized. Mangrove mortality commences when growth suppression ($f \cdot C < 0.5$) of each mangrove age and size class continues for 5 consecutive years, triggering a self-thinning process[92]. The number of trees is reduced until their suppressed growth terminates or no vegetation is left in the cell. After mortality and at the beginning of every new ecological year, colonization restarts and cells with suitable growth ($f \cdot C > 0.5$) are allowed to have new seedlings.

Overall, the vegetation model calculates several vegetation parameters, including the sizes and densities of vegetation objects (i.e. stems and roots), which are used to calculate hydraulic resistance in Delft3D so that the effects of mangroves on tidal flow and consequently sediment transport are accounted for[92].

## Idealized landscape settings

Estuarine systems usually vary with characteristics depending on various factors, including geomorphology, the evolutionary stage, hydrology and salinity or combinations of the above[96]. New Zealand mangroves usually colonize barrier-enclosed or headland-enclosed estuaries with inlets restricted by rocky headlands, typically with multiple catchments that are small relative to estuary surface area[21]. Here we simulate an idealized back-barrier fluvial-tidal basin to represent these typical estuarine systems in the North Island, New Zealand, most of which are characterized by similar spatial scales, similar historical vegetation expansion trends and similar sedimentation accumulation processes (Fig. 1). The model domain consists of a 4 km by 2 km estuarine basin enclosed by two non-erodible barriers, connected to the open coast with a 2 km by 2 km rectangular offshore area (Fig. 7). The grid resolution is set to 15 m by 15 m to allow evolution of channel networks and changes of mangrove forests[92,97–99]. The initial bed elevation in the basin is set to 1.5 m below mean sea level, while the offshore area attains a sharp slope from −1.5 in the inlet to −100 m at the offshore boundary (Fig. 7b). This large depth avoids shallowing by sedimentation outside the inlet, as wind and waves are not incorporated for computational efficiency. We include three fluvial inflows in our model from different directions, which is typical of estuarine systems on the North Island, New Zealand (Fig. 1b–d). Following an assessment of the hydraulic geometry of 73 New Zealand river reaches, the average river width is around 28 m (see Supplementary Data 2.xlsx)[100] and we thus set the river width to twice the cell size (i.e. 30 m) (Fig. 7). The chosen river width is consistent with the empirical relations between river discharge and river width described in the next section (Eq. 5).

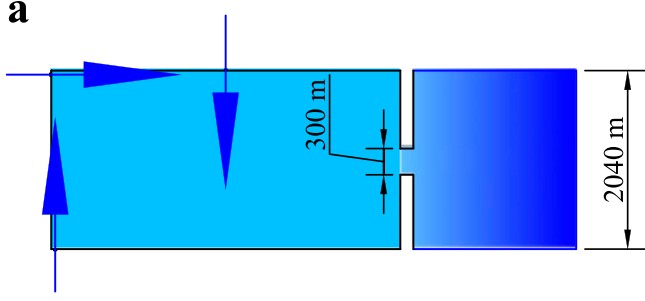

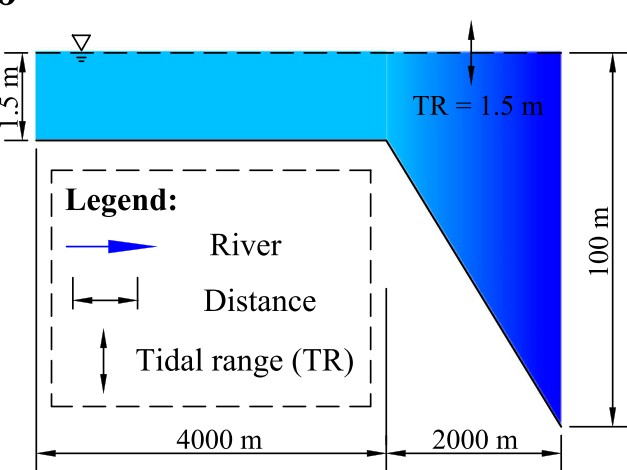

**Fig. 7 | Model layout comprising size, initial bathymetry and boundaries.** Plan view of model domain with (**a**) three river inputs; (**b**) cross-section view along the domain.

## Hydrodynamic forcing

We use the NIWA tide model to calculate the annual mean tidal range at Whangapoua, Whangamatā and Wharekawa estuaries (https://niwa.co.nz, also see Supplementary Data 4.xlsx); the mean tidal range is around 1.5 m which is consistent with observations[101,102]. We apply an M2 tidal cycle with a 1.5-m range and 30-deg/hr frequency at the seaward boundary[103]. Both the northern- and southern-seaward boundaries are set as Neumann conditions. The river discharge is set to 18 m³/s based on the average of the data from 73 New Zealand river reaches (Fig. S6, also see Supplementary Data 2.xlsx)[100]. This value is similar as that derived from another dataset which contains both river discharge and the corresponding suspended sediment yields based on nearly 150 observational sites in North Island, New Zealand (Supplementary Data 1.xlsx)[22]. Furthermore, the selected river conditions are consistent with empirical relations between river width and flow discharge as[104,105]

$$W = 7.2Q^{0.5} \tag{5}$$

where $W$ is the river width (m) and $Q$ is the flow discharge (m³/s). When applying a flow discharge $Q$ of 18 m³/s in Eq. 5, the calculated river width is about 30.5 m, which is nearly the same as our predefined value (i.e. 30 m) based on fluvial geometry data from Jowett[100].

## Sediment supply settings

The model accounts for both sand and mud transport. The median grain size of sand is set to 250 µm consistent with previous observations[49]. At the flow boundaries (river and sea), we apply the concept of equilibrium sediment concentration for sand such that the amount of sand imported into/exported from the system depends on the flow velocity. This setup allows model boundaries to adapt to local flow conditions such that little deposition and erosion can occur near

the boundaries[78]. Mud supply is controlled to represent upstream land-use changes. Catchment deforestation, agriculture and mining have enhanced soil erosion. Sediment is then washed into rivers and subsequently deposited in estuarine systems[21,24]. To model estuarine infilling for different types of upstream land-use and variations in catchment characteristics, low, intermediate and high concentrations of cohesive sediment are supplied at the river boundaries in different simulation periods. Based on sensitivity tests, we use 5 mg/L as the sediment concentration before human disturbances and 15 mg/L when simulating bio-geomorphic change under enhanced sediment inputs. These sediment settings provide annual sediment yields that are within the range reported for New Zealand systems (Fig. S7a), and result in realistic non-dimensional catchment sediment yields (Figs. 5 and S1b) and mangrove expansion rates (Fig. 3).

## Mangrove species and their characteristics

*Avicennia marina* is the main mangrove species on the North Island of New Zealand. We therefore set up vegetation properties based on *Avicennia marina* to represent local mangrove species[55]. Although *Avicennia marina* can grow as high as 10 m[89], New Zealand mangrove trees are relatively small with varying height among different estuarine systems[13]. Observations of New Zealand mangrove dimensions indicate that the maximum mangrove height rarely exceeds 4 m to 6 m[106,107]. Thus, in the model, we set the maximum vegetation height and diameter to 3.2 m and 0.18 m following observation from the Whangapoua estuary (Table S1)[55]. The height of mangrove pneumatophores in New Zealand is typically around 5 to 25 cm[13]. We therefore set a fixed height (i.e. 10 cm) for mangrove pneumatophores but vary the number of pneumatophores as a function of mangrove tree size following previous mangrove modeling studies[42,76,92].

## Model scenario setup

The models were initiated by a 200-year spinup period to create an initial morphology with a stable cross-sectional inlet depth (Fig. S8). In

the pre-disturbance period, low mud supply (using 5 mg/L) is prescribed at the river boundaries to simulate conditions with minor human intervention, representing the situation before European settlement in New Zealand. During the disturbance period, we used a high mud supply (15 mg/L) at the river boundaries, representing the situation after European settlement (with deforestation and agricultural practices in the catchments). Impacts of different management approaches were then tested (Fig. 8), including mangrove removal according to different coverage reductions (i.e. 25%, 50%, and 100% mangrove removal). Mangrove removal was conducted by completely removing both stems and roots, following local studies documenting mangrove removal practices in New Zealand[38,108]. A sensitivity analysis was carried out to evaluate the impacts of root persistence on estuarine infilling processes as shown in Fig. S12. Leaving roots in place for longer periods of time initially constrains estuarine infilling, however, the subsequent disappearance of roots (due to decomposition) then accelerates estuarine infilling and the formation of muddy areas. Under short root persistence periods such as 2 or 5 years based on field observation[49,57], estuarine infilling and muddy area development follow similar trends as the reference in which both stems and roots were removed completely. Thus, the influence of root persistence on general trends in estuarine dynamics is limited. We also conducted scenarios in which mud supply was reduced to pre-disturbance and intermediate levels, after different disturbance durations. Overall, this allowed us to evaluate the effects of contrasting management approaches on estuarine landscape development.

## Estuarine infilling data specification

The modeled estuarine infilling patterns among different scenarios were compared to previous published data from different estuarine systems in New Zealand (Fig. 5; details can be seen in Table S2). For these New Zealand systems, tidal prism (TP) is calculated from spring low and high tide volumes[96]. The sediment yield (SY) is estimated through use of the USGS SPARROW model based on data from the NZ

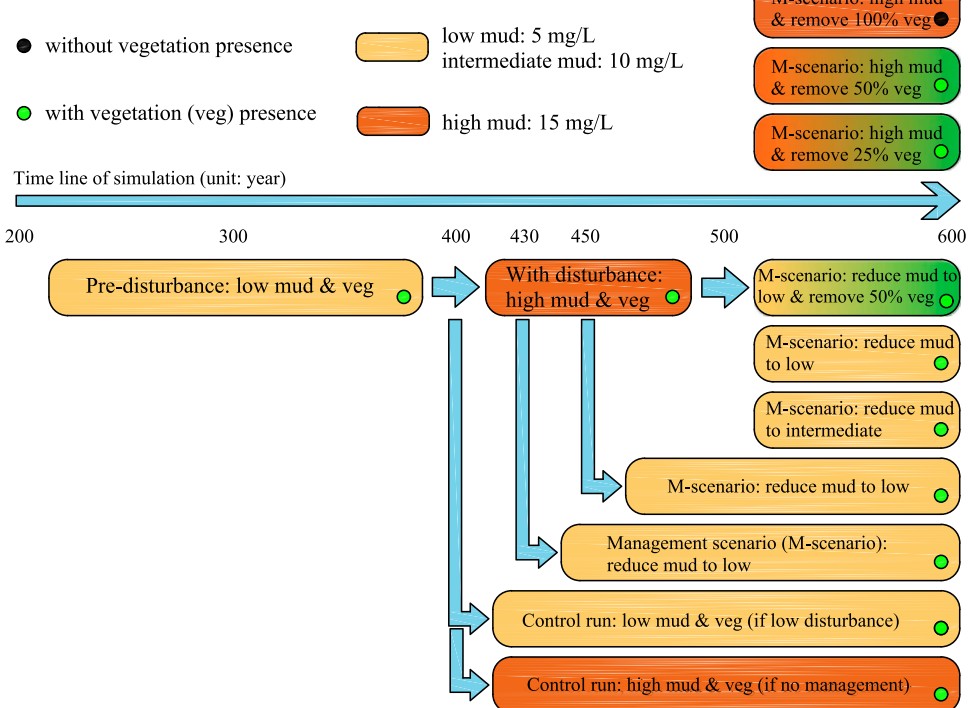

**Fig. 8 | Flow chart of the design of model simulations.** Light- and dark-brown colors are used to highlight the sediment conditions of each river inflow, with low/intermediate mud and with high mud input. Vegetation presence is indicated by green dots in the bottom-right of each box, while black dots indicate vegetation absence. Additional control runs with different amounts of mud input in the absence of vegetation are simulated but not displayed in the above diagram.

River Environment Classification calibrated by Elliott et al. [109] and Hicks et al. [110] with a regression fit ($r^2$) of 0.925. The sediment yields of Whangapoua, Wharekawa and Whangamatā estuaries are derived from NIWA sediment model[111]. The relative basin area above mean sea level ($A_{MSL}/A_{tot}$) is defined as the ratio between the area of intertidal habitat above mean sea level and estuary surface during high tide. The data source of these surface measurements is based on NIWA Estuarine Environment Classification Database[112].

## Data availability
The data regarding mean discharge and width of New Zealand rivers are available as supplementary material (Supplementary Data 1.xlsx and Supplementary Data 2.xlsx). Estuarine infilling data of New Zealand estuaries consisting of annual sediment yield and tidal prism are summarized in Table S2 (see Supplementary Information 1.pdf) and Supplementary Data 3.xlsx. Delft3D is an open-source code available online (at https://oss.deltares.nl).

## Code availability
The dynamic vegetation code with a representative model set-up is available at the repository Zenodo (https://doi.org/10.5281/zenodo.8356151)[113].

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

## Acknowledgements

D.X acknowledges the financial support from the Department of Physical Geography, Utrecht University. K.R.B. acknowledges funding from the New Zealand Ministry of Business Innovation and Employment (MBIE) Future Coasts Aotearoa Contract (grant no. C01X2107). M.G.K. acknowledges funding from ERC Consolidator (grant no. 647570). B.v.M. acknowledges funding from the Natural Environment Research Council (grant no. NE/V012800/1). We acknowledge the many Māori tribes who are the traditional custodians of mangrove-dominated estuaries in Aotearoa New Zealand.

## Author contributions

D.X., C.S., and B.v.M. conceptualized the study, designed the methodology, performed the analyses of results. D.X. ran the simulations and led the writing of the paper. M.G.K., K.R.B., G.C., and S.H. performed the analyses of results. D.X., B.v.M., C.S., M.G.K., K.R.B., G.C., and S.H. wrote the paper. All authors provided comments on the data processing and substantially contributed to the drafts.

## Competing interests

The authors declare no competing interests.
