## [Peer Review File · Nature Communications]

Mangrove removal exacerbates estuarine infilling through landscape-scale bio-morphodynamic feedbacksREVIEWER COMMENTS

Reviewer #1 (Remarks to the Author):

This manuscript describes an excellent study on the changes in estuary sediment dynamics and infilling with human impacts to coastal watersheds and removal of mangroves. This study is very relevant to local/regional coastal management, where restoration involves removing mangroves with the goal of reducing turbidity and infilling of the estuary. The management strategy of removing mangroves is in contrast to much of the scientific literature that documents their ability to trap sediments and improve water quality. The study findings provide further support for this idea.

Overall, the study is very well written and comprehensive. For a broad Nature readership, I would suggest reconciling the ideas in the first paragraph (i.e., "the problem" of increased sediment supply to the coast due to land-use changes in the watershed) with the predominant literature for many other coastal areas, a reduced sediment supply due to river management, sea-level rise and a loss of coastal wetland area.

The main findings of this study that watershed sediment supply via erosion has a much stronger control on estuarine sediment input than mangrove establishment are important, but not particularly surprising.

Reviewer #2 (Remarks to the Author):

In this manuscript, the authors use a morphodynamic model of a mangrove estuary to explore complex feedbacks between sediment supply and vegetation in the context of controversial management programmes intended to address estuarine infilling through mangrove removal.

This work represents a novel and exciting contribution to the field of human–coastal systems – and is also an excellent example of how a deliberately simplified morphodynamic model can be used as a powerful exploratory tool for physical insight. The manuscript itself is well-written and well-organised. The logic of the exercise is clear, and impart the momentum of that clarity into the conclusions.

My primary suggestion that is that the authors revisit their results sections (L182, L220 & 253) to make sure the flow through each is as straightforward as they can be. I just urge them to watch the number of reversals they make (e.g., "However,..." "Conversely,...") because it's easy for a reader to lose their grip on the comparative scenarios in question. Similarly, in the section at L253, I suggest that rather than use the nondimensional term A_{msl}/A_{tot} in the text after L256, use the explanatory term ("relative basin area") after its first invocation. This is just for readability – consider the difference at L256: "Large relative basin areas resembled infilled estuaries with..." The figures already contain all the technical elements; the text then becomes that much easier to absorb.

Finally, I have a minor suggestion regarding the title. I think the forest/trees expression works well in

the body of the document, but I don't think it works as a hook for the title. Given the authors' existing title, perhaps the following would better serve: "How focus on local bio-morphodynamic feedbacks compromises management of mangrove systems at the estuary scale"

I look forward to seeing this work in print, and am glad to have had the opportunity to read it at the review stage.

Reviewer #3 (Remarks to the Author):

This is an interesting and well written paper modelling the importance of two drivers of estuarine change in New Zealand- mud delivery to the estuary from the catchment, and the management of mangroves. The finding that catchment sediment delivery is a dominant control on estuarine infilling and mangrove expansion is well made, if unsurprising. The surprising result is that mangrove removal is modelled to have the opposite effect of that intended by this management intervention. By removing mangroves, tidal water is more likely to deliver sediment to the broader intertidal platform, promoting estuarine infill and exacerbating the problem of mud and mangrove management. The logical case for this outcome is presented- that the resistance provided by mangrove vegetative structure promotes sedimentation on levees and enhances flow in tidal channels. The idea is sufficiently novel to justify publication, and the result is highly relevant to one of the big coastal management issues in New Zealand, and possibly other locations impacted by increased sedimentation following catchment modification.

I have some questions relating to the parameterisation of the models which the authors may wish to consider. As acknowledged in the manuscript, experimental evidence of the processes being described in the model is limited, given the challenge in measuring these processes at the temporal and spatial scales over which they operate. Given this, it is worth considering some evidence which might question some elements of the model

(i) I was surprised that ongoing sedimentation doesn't restrict mangrove area as accommodation is filled. Perhaps there is insufficient time for this in the model run, but the elevation of intertidal areas above that suitable for mangrove colonisation can happen quite quickly (see Woodroffe's work on the Holocene stratigraphy of northern Australian estuaries for an excellent treatment: Woodroffe et al. 2016). Mangrove will only occur between MSL and MHW- are the upper bounds of mangrove elevation correctly represented in the model?

(ii) Mangrove removal would most probably leave the root structures in place, and so it may not be the case that this influences hydraulic resistance to the extent proposed in the model. Also, one probable short-term impact of mangrove removal is likely to be an increase in accommodation, due to below-ground root collapse (See Cahoon et al. 2003; Lang'at et al. 2014).

(iii) Mangroves reduce accommodation through vertical root growth, which can be the dominant contributor to elevation gain (see McKee et al. 2007 in a different context, but making this point).

One final point is that estuarine environments across the globe are subject to increasing accommodation due to accelerating sea-level rise. Over coming decades this may become a counter-

balancing effect on catchment sediment delivery (which is surely a pulse event). In many locations mangroves are expanding due to increased accommodation (moving landward into saltmarshes: Kelleway et al 2017), rather than decreased accommodation. This broader context could be considered in the discussion.

Minor points

Fig 1. Make clear that mangrove coverage (%) refers to percentage of estuarine area (a reader may assume you mean percentage increase against original extent)

Fig 3. Could the x107 relating to the units of the y-axis appear in the axis title? I missed it the first time

Line 120. Consider citing the pioneering work of Bruce Thom here- your point about mangroves being a more passive element in estuarine geomorphology was emphatically made in his early work (e.g. Thom 1967).

Line 320. I couldn't find Wu et al. 2001 or Montgomery et al. 2022 in the reference list. These are very important references in the context of your argument, and I would have liked to refer to them.

Figure 6. You refer to "positive feedbacks" in this diagram when I think you mean relationships. For example, the relationship between soil erosion and sediment delivery to the coast is not a "positive feedback", because sediment delivery to the coast has no influence on soil erosion.

References

Cahoon, D. R., Hensel, P., Rybczyk, J., McKee, K. L., Proffitt, C. E., & Perez, B. C. (2003). Mass tree mortality leads to mangrove peat collapse at Bay Islands, Honduras after Hurricane Mitch. *Journal of ecology*, 91(6), 1093-1105.

Kelleway, J. J., Cavanaugh, K., Rogers, K., Feller, I. C., Ens, E., Doughty, C., & Saintilan, N. (2017). Review of the ecosystem service implications of mangrove encroachment into salt marshes. *Global Change Biology*, 23(10), 3967-3983.

Lang'at, J. K. S., Kairo, J. G., Mencuccini, M., Bouillon, S., Skov, M. W., Waldron, S., & Huxham, M. (2014). Rapid losses of surface elevation following tree girdling and cutting in tropical mangroves. *Plos one*, 9(9), e107868.

McKee, K. L., Cahoon, D. R., & Feller, I. C. (2007). Caribbean mangroves adjust to rising sea level through biotic controls on change in soil elevation. *Global Ecology and Biogeography*, 16(5), 545-556.

Thom, B. G. (1967). Mangrove ecology and deltaic geomorphology: Tabasco, Mexico. *The Journal of Ecology*, 301-343.

Woodroffe, C. D., Rogers, K., McKee, K. L., Lovelock, C. E., Mendelssohn, I. A., & Saintilan, N. (2016). Mangrove sedimentation and response to relative sea-level rise. *Annual review of marine science*, 8, 243-266

COLOR KEY

Black: Reviewers' comments (**Numbered**, e.g., **1.1**)

Blue: Authors' responses (**Numbered**, e.g., **R1.1**) and text from the previous manuscript

~~Red: Text removed from the previous manuscript~~

Gold: New text incorporated in the revised manuscript

The line numbers mentioned here refer to the Word version with tracked changes (e.g., Line 1).

REVIEWER COMMENTS

Reviewer #1 (Remarks to the Author):

This manuscript describes an excellent study on the changes in estuary sediment dynamics and infilling with human impacts to coastal watersheds and removal of mangroves. This study is very relevant to local/regional coastal management, where restoration involves removing mangroves with the goal of reducing turbidity and infilling of the estuary. The management strategy of removing mangroves is in contrast to much of the scientific literature that documents their ability to trap sediments and improve water quality. The study findings provide further support for this idea.

We thank the reviewer for the nice words and constructive comments.

1.1

Overall, the study is very well written and comprehensive. For a broad Nature readership, I would suggest reconciling the ideas in the first paragraph (i.e., “the problem” of increased sediment supply to the coast due to land-use changes in the watershed) with the predominant literature for many other coastal areas, a reduced sediment supply due to river management, sea-level rise and a loss of coastal wetland area.

R1.1

Thanks for the suggestion. We agree with the reviewer that the existing literature predominantly focusses on the impact of insufficient sediment supply, sea-level rise and the consequent loss of coastal wetlands, while here we deal with sediment-rich systems showing wetland expansion. We now added an additional statement at the very beginning to highlight the general issue of changing coastal sediment supply. We note the contrasting situations of either a lack or surplus of sediment availability and implications for wetland systems.

Lines 62-71:

Coastal wetlands are crucially important but under pressure from a range of different drivers including changing sediment availability. The loss of coastal wetlands due to the shortage of riverine sediment supply (driven by human activities such as dam construction) in combination with sea-level rise has received significant attention in recent decades (Kirwan and Megonigal 2013, Lovelock et al. 2015), while relatively less attention has been devoted to coastal wetlands with excessive sediment supply (Kirwan et al. 2011, Nienhuis et al. 2020). Over the last few centuries, land-use changes and coastal development have markedly increased sediment supply towards to the coast in many parts of the world (Syvitski et al., 2005; Vanmaercke et al., 2015; Dethier et al., 2022), leading to substantial physical transformations of coastal landscapes (Nienhuis et al., 2020) and ecosystems (Kirwan et al.,

2011; Suyadi et al., 2019).

1.2

The main findings of this study that watershed sediment supply via erosion has a much stronger control on estuarine sediment input than mangrove establishment are important, but not particularly surprising.

R1.2

The novelty of our work is indeed not that greater sediment input causes more sedimentation in estuaries (which we agree is an obvious outcome), rather that mangroves counterintuitively decrease this estuarine infilling. Thus, our research goes much further by unravelling how bio-morphodynamic feedbacks operate at different scales and can cause counterintuitive landscape-scale responses. The surprising result is that mangrove removal increases estuarine infilling, and thus aggravates the problem of muddification, as also mentioned by the third reviewer. In response to the above comment and the feedback of reviewer 3, we now provide clearer information on the processes leading to this unexpected result.

In the revised version, we have added new evidence showing how vegetation removal changes the fundamental nature of channels by affecting flow direction and levee formation. We further provide additional mechanistic insights on how these processes depend on the dominant hydrodynamics within a specific channel tributary (e.g., tides or river flow). To do this, we categorize channels into river-dominated or tide-dominated by evaluating the direction of the flow during flood and ebb (Fig. R1). The absence of a landward (up-estuary) flow during the flood tide is considered indicative of river flow dominance. River-dominated channels receive sediment directly from the catchment whereas tide-dominated sections of the channel network that are not directly connected to riverine inputs receive sediment from the lower estuary through tidal processes.

Figure R1. (Figure S9) Distribution of tide-dominated and river-dominated channels based on the scenario of Fig. 2d in the main text. When the flow is directed seaward even during the flood period, the channel is considered as river-dominated and marked in green (see example in Fig. R2). In contrast, when the flow is directed landward during the flood period and flow reversals occur, the channel is considered to be tide-dominated and marked in blue (see example in Fig. R3).

Our original manuscript showed that the presence of vegetation and the enhanced development of levees impede sediment supply to the mangrove platform (Figs. 4 & S4). Two new figures have been incorporated in the supplementary material to illustrate the process and isolate where within the estuary vegetation impacts are most profound. Specifically, we show the relationship between water level and velocity along a river-dominated channel and a tide-dominated channel, respectively (Figs. R2-3). In river-dominated channels, the presence of vegetation amplifies seaward-directed flows (Fig. R2c-e). As such, vegetation increases river discharge dominance and can even prevent landward (up-estuary) flow during flood tide at the most downstream location (Fig. R2e, point 3). Levees are much more developed along these river-dominated channels (Figure R1) thus indicating localized sedimentation. Levee formation is clearly a function of the distance from the sediment source, with levee elevations in upstream locations approaching the level of the high-water (Fig. R2c&d, point 1 and 2; levee elevations are indicated by the horizontal dashed lines). The elevation of levees decreases rapidly in the downstream direction, especially when mangrove vegetation is removed highlighting again the role of vegetation in levee development (Fig. R2c,e).

In the case of the tide-dominated channel, vegetation creates a stronger current predominantly during the flood stage of the tide, in comparison to the scenario without vegetation (Fig. R3c-h). This is especially the case in the more landward part of the tide-dominated channel and is accompanied here with increased levee elevation despite being further away from the main sediment source (Fig. R3c, point1).

We now have included these three figures as supporting information and extend the discussion to clarify the effect of vegetation on estuarine landscape development:

Lines 416-435:

Furthermore, the presence of vegetation has previously been found to promote levee growth in fluvial-tidal environments (Boechat Albernaz et al. 2020). Levee development is of critical importance as levees not only store sediment locally directly adjacent to channels, but also because they influence channel hydro-sedimentary processes and the delivery of sediment to the vegetated platform (Temmerman et al. 2004, Fagherazzi et al. 2012, Boechat Albernaz et al. 2020). Our research reveals that vegetation effects on levee formation and associated channel flow depend on along-channel location and whether the channels are dominated by tides or river flow (Figs. S9-11). For river-dominated channels, vegetation strengthens seaward directed flows and can even suppress flow reversals during the flood tide (Fig. S10). As such, river-dominated channels flanked by vegetation become more effective conduits for water and sediment transport, with implications for estuarine sediment budgets. For channels driven by river flow, levee height generally decreases with increasing distance from the sediment source. Mangroves are found to promote levee formation but only at downstream locations. In contrast, for tide-dominated channels, vegetation mainly enhances flood currents and contributes to levee elevation but here only at the more upstream locations (Fig. S11). Clearly, vegetation effects on ~~landscape-scale~~ sedimentation patterns and changes following vegetation removal are not simply determined by the lateral distance from a channel, but can spatially vary and depend on the dominant drivers of individual channel distributaries. Mutual interactions between vegetation, vegetation-influenced landforms and sedimentation patterns are extremely complex and deserve further attention ~~with large~~ given the significant implications for management and ecosystem resilience.

Figure R2. (Figure S10) Relation between water level and velocity within a river-dominated channel, with and without mangrove presence. a-b) Locations selected for comparisons. c-e) Tidal stage plots for the different locations along the channel. The horizontal green dashed lines indicate the vegetation elevation near the channel, and the elevation for the same area without vegetation is marked by a horizontal black dashed line f-h) Velocity vector at the selected points along the channel. The blue dashed lines delineate the channel band.

Figure R3. (Figure S11) Relation between water level and velocity within a tide-dominated channel, with and without mangrove presence. a-b) Locations selected for comparisons. c-e) Tidal stage plots for the different locations along the channel. The horizontal green dashed lines indicate the vegetation elevation near the channel, and the elevation for the same area without vegetation is marked by a horizontal black dashed line. f-h) Velocity vector at the selected points along the channel. The blue dashed lines delineate the channel band.

As the reviewer highlights, sediment supply from the catchment plays a key role in determining estuarine mud infilling, thereby implying that reducing mud input would be an effective approach to limit estuarine muddification, but the complex/spatially variable bio-morphodynamic feedbacks play a central role in how the impact is expressed. In addition, we highlight the critical role of the timing of fluvial mud reduction in achieving this goal. Taking early action to reduce mud supply from upstream can effectively mitigate estuarine infilling, whereas delayed responses may not yield desired outcomes (Lines 340-343).

Reviewer #2 (Remarks to the Author):

In this manuscript, the authors use a morphodynamic model of a mangrove estuary to explore complex feedbacks between sediment supply and vegetation in the context of controversial management programmes intended to address estuarine infilling through mangrove removal.

This work represents a novel and exciting contribution to the field of human–coastal systems – and is also an excellent example of how a deliberately simplified morphodynamic model can be used as a powerful exploratory tool for physical insight. The manuscript itself is well-written and well-organised. The logic of the exercise is clear, and impart the momentum of that clarity into the conclusions.

We very much appreciate the reviewer's positive evaluation of our study. Thank you for your comments.

2.1

My primary suggestion that is that the authors revisit their results sections (L182, L220 & 253) to make sure the flow through each is as straightforward as they can be. I just urge them to watch the number of reversals they make (e.g., "However,..." "Conversely,...") because it's easy for a reader to lose their grip on the comparative scenarios in question. Similarly, in the section at L253, I suggest that rather than use the nondimensional term A_{msl}/A_{tot} in the text after L256, use the explanatory term ("relative basin area") after its first invocation. This is just for readability – consider the difference at L256: "Large relative basin areas resembled infilled estuaries with..." The figures already contain all the technical elements; the text then becomes that much easier to absorb.

R2.1

We want to thank the reviewer for providing these insights. We have now made the following changes to improve readability.

First, we rephrased the results sections by reducing the number of transitional words or phrases used in the text.

Lines 212-216:

A current management practice in New Zealand estuaries involved localized mangrove clearance but model simulations indicated that this intervention did not limit ongoing muddification. ~~In contrast,~~ Mangrove removal *in fact* enhanced estuarine infilling and resulted in a larger portion of the estuary to consist of muddy substrate (Fig. 3a&b). These effects were exacerbated with increasing levels of mangrove removal.

Lines 240-245:

As a consequence, the overall area with muddy substrates in the estuary might not reduce, but rather increase through mangrove removal (see also Fig. 3b). ~~Conversely, model results showed that~~ Under reduced mud supply (blue violins in Fig. 4), *model results showed that* both channelized and unchannelized areas would retain more similar bed elevations, mud thicknesses and sedimentation rates to those observed at the pre-disturbance stage.

Then, in the result section ‘Estuarine landscape trajectories’, we rephrased the text to avoid overuse of transitional words or phrases and replaced the nondimensional term A_{MSL}/A_{tot} in the text with the explanatory term (‘relative basin area a_{MSL} ’) to improve its readability.

Lines 263-283

To relate basin response directly to catchment management and disturbance, we evaluated the relative basin area above mean sea level (a_{MSL}) (A_{MSL}/A_{tot}) against the non-dimensional catchment sediment yield (SY/TP) (Fig. 5). Large ~~values of A_{MSL}/A_{tot}~~ *basin areas a_{MSL}* resembled infilled estuaries with extensive potential mangrove habitats, while small ~~values~~ *areas a_{MSL}* corresponded to unfilled systems dominated by unvegetated flats and subtidal areas. Model results followed previously described estuarine infilling patterns driven by fluvial sediment yields found in New Zealand (Swales et al., 2021). We found that estuarine landscapes deviated strongly from their pre-disturbed state when an increased mud supply was maintained, causing ~~A_{MSL}/A_{tot}~~ *relative basin area a_{MSL}* to exceed 0.5 (Fig. 5 *). This implied that more than half of the estuarine area was suitable for mangrove growth. ~~In contrast, A_{MSL}/A_{tot}~~ *Relative basin area a_{MSL}* remained below 0.2 for the simulated estuary with continued low mud supply (\square). Removal of mangroves slightly enhanced the formation of upper intertidal area and thus unexpectedly facilitated the creation of new mangrove habitat (\diamond). Model simulations showed that estuarine infilling and intertidal area development could be limited through reductions in sediment supply, but the magnitude and especially timing of such reductions were critical as the dependency of ~~A_{MSL}/A_{tot}~~ *basin areas a_{MSL}* on sediment supply was not linear. For example, when sediment yield was reduced back to the pre-disturbed condition after 100 years, ~~A_{MSL}/A_{tot}~~ *basin area a_{MSL}* increased more rapidly during subsequent lower sediment yield (from 0.27 to 0.36 between 500 and 600 years; Δ) than when sediment yield was restored after 50 years (from 0.17 to 0.26 between 450 and 600 years; O). This highlighted that estuarine landscapes were not only modified throughout the disturbance period, but that also longer-term effects on landscape trajectories were controlled by disturbance duration.

2.2

Finally, I have a minor suggestion regarding the title. I think the forest/trees expression works well in the body of the document, but I don't think it works as a hook for the title. Given the authors' existing title, perhaps the following would better serve: "How focus on local bio-morphodynamic feedbacks compromises management of mangrove systems at the estuary scale"

R2.2

We understand that the phrase ‘Missing the forest for trees’ in the previous title was perhaps not clear. We have now removed this part of the title and made some modifications to create an effective title that conveys our key message “*Mangrove removal exacerbates estuarine infilling through landscape-scale bio-morphodynamic feedbacks*”.

I look forward to seeing this work in print, and am glad to have had the opportunity to read it at the review stage.

We want to thank the reviewer again for the compliments and for the constructive feedback on our manuscript. We hope the revisions made on the text, figures and the title above have improved the narrative of this research.

Reviewer #3 (Remarks to the Author):

This is an interesting and well written paper modelling the importance of two drivers of estuarine change in New Zealand- mud delivery to the estuary from the catchment, and the management of mangroves. The finding that catchment sediment delivery is a dominant control on estuarine infilling and mangrove expansion is well made, if unsurprising. The surprising result is that mangrove removal is modelled to have the opposite effect of that intended by this management intervention. By removing mangroves, tidal water is more likely to deliver sediment to the broader intertidal platform, promoting estuarine infill and exacerbating the problem of mud and mangrove management. The logical case for this outcome is presented- that the resistance provided by mangrove vegetative structure promotes sedimentation on levees and enhances flow in tidal channels. The idea is sufficiently novel to justify publication, and the result is highly relevant to one of the big coastal management issues in New Zealand, and possibly other locations impacted by increased sedimentation following catchment modification.

I have some questions relating to the parameterisation of the models which the authors may wish to consider. As acknowledged in the manuscript, experimental evidence of the processes being described in the model is limited, given the challenge in measuring these processes at the temporal and spatial scales over which they operate. Given this, it is worth considering some evidence which might question some elements of the model.

We thank the reviewer for the positive evaluation and for the constructive feedback on our manuscript. In the revised manuscript we have now addressed the raised issues and concerns highlighted by the reviewer and we provide a detailed clarification below.

To provide more experimental evidence for comparison with our model simulations, we have identified a recent study by Schepers et al., 2019, which shows through a field experiment that sedimentation in the interior of a freshwater tidal marsh can increase following vegetation removal. As the reviewer highlights, there is a challenge in terms of scales and this field experiment was carried out over a much smaller spatial scale and with sedimentation rates evaluated over much smaller temporal scales. Despite the difference in scales, and the different type of environment (mangrove vs freshwater marsh), their findings are supportive of our modelling and we therefore included this in the revised manuscript:

Lines 406-414:

For a freshwater tidal marsh, vegetation removal ~~was studied by Temmerman et al. (2012) and, in agreement with our study, they found that vegetation clearance increased flow velocities on the platform while reducing channel flow velocities~~ *experiments have shown that vegetation clearance reduced channel flow velocities while increasing flow velocities on the platform (Temmerman et al., 2012). They suggest that large-scale vegetation die-off would result in decreased platform sedimentation rates while our simulations of mangrove removal indicate that accretion in unchanneled areas increases because of enhanced sediment delivery. This causes a spatial redistribution of sediment with the potential of enhanced sedimentation in the inner marsh (Schepers et al. 2020) which is in agreement with our model findings.*

3.1

(i) I was surprised that ongoing sedimentation doesn't restrict mangrove area as accommodation is filled. Perhaps there is insufficient time for this in the model run, but the elevation of intertidal areas above that suitable for mangrove colonisation can happen quite quickly (see Woodroffe's work on the Holocene stratigraphy of northern Australian estuaries for an excellent treatment: Woodroffe et al. 2016). Mangrove will only occur between MSL and MHW- are the upper bounds of mangrove elevation correctly represented in the model?

R3.1

We agree with the reviewer and Woodroffe et al. (2016), that mangrove distribution is confined to the area between MSL and MHW, and our dynamic vegetation model includes this behaviour, although we allow some time at sub-optimal conditions at the top of the tidal range before the mangroves die off. Below we explain how this process is incorporated and then show the changes of vegetation characteristics as a function of bed elevation.

Our model includes colonization, growth and mortality of mangrove trees, which are mainly controlled by two factors: the fitness function (f) and the competition stress factor (C) (Fig. R4). The fitness function is based on the premise that an optimal relative hydroperiod (P) exists (i.e. $f = 1$ when $P = 0.25$) for mangroves, and deviations from this optimal value (P either larger or smaller) result in reduced growth (Fig. R4a). Meanwhile, the growth of mangrove forests is further constrained by the availability of resources, which is introduced as a competition stress factor (C) that evaluates the competition between one specific tree and surrounding mangrove vegetation (Fig. R4b).

Figure R4. Growth control factors used in the vegetation model (van Maanen et al. 2015). (a) Fitness function, characterized by an optimal relative hydroperiod ($P = 0.25$). $P > 0.6$ implies over-inundation and $P = 0$ indicates the non-flooding situation. (b) Competition stress factor, representing the competition between trees as neighbouring trees have to share resources.

The value of $f \cdot C$ not only determines the vegetation growth rate, but also acts as an index evaluating growth conditions. When $f \cdot C < 0.5$, i.e. growth rate is smaller than 50% of the optimal condition, mangrove growth is considered to be significantly constrained. The mortality of mangrove trees is

identified when consecutive depressional growth occurs. In our model, mangroves die when vegetation growth is suppressed for 5 consecutive years.

Under the above settings, after initial colonization and as the accommodation space fills due to sedimentation, the relative hydroperiod (P) gradually decreases to 0. Consequently, the fitness function (f) initially increases to 1, providing optimal growth conditions for vegetation, but then reduces to 0.5 due to reduced flooding. When vegetation already exists in a cell, the competition stress factor is generally below 1. Thus, a fitness function of 0.5, combined with a competition stress factor lower than 1, yields a value of $f \cdot C$ smaller than 0.5. If this sub-optimal condition persists for 5 consecutive years, mangrove trees start dying, leading to a reduction in vegetation density until no mangrove trees remain. From this perspective, our model does include the process whereby ongoing sedimentation constrains vegetation growth.

To illustrate that our model includes the process of constrained growth caused by sedimentation, we extracted the fitness function, vegetation density and bed level data from all vegetated areas in the simulation shown in Fig. 2d. We here present the changes in fitness function and vegetation density ($1/m$) (defined as the number of vegetation objects ($1/m^2$) \times vegetation diameter (m)) as a function of bed level (Fig. R5). As the bed level increases towards the high-water level, the fitness function decreases (Figs. R4a&R5a). Consequently, mangrove growth is constrained and results in lower vegetation densities within the upper intertidal area (Fig. R5b).

Figure R5. Distribution of fitness function and vegetation density as a function of bed level. Sample points in a and b are based on the vegetated cells extracted from Fig. 2d in our main text. The cells previously occupied by vegetation but later become unvegetated are not shown in the above plots. Solid lines show the mean value, while the shaded area represents the 95% confidence interval.

To clarify the process that sedimentation eventually constrains mangrove growth, we have now included more details in the method section:

Lines 504-511:

As mangroves mainly occupy areas *above between* mean water level *and mean high water* (Chapman, 1976; Woodroffe et al. 2016) and seedling establishment is hindered under larger bed shear stresses induced by currents/waves (Balke et al., 2011), we assign an initial vegetation density to the cells with relative hydroperiod ranging between 0 and 0.5, and bed shear stress below 0.2 N/m². The initial seedling density is set to 3000 individuals/ ha following van Maanen et al. (2015). *Infilling of accommodation space due to sedimentation can suppress the growth of mangroves and result in a lower vegetation density if the upper limit of mangrove elevation is being reached* (Xie et al., 2022).

3.2

(ii) Mangrove removal would most probably leave the root structures in place, and so it may not be the case that this influences hydraulic resistance to the extent proposed in the model.

R3.2

Before setting up our models, we shared the same concern as the reviewer regarding the potential influence of mangrove root structure on hydraulic resistance. However, the approach we adopted in this research, in which we remove both mangrove stems and aboveground root structures, is informed by previous reports and scientific research. The following study from Swales et al. (2009) describes mangrove removal activities conducted in New Zealand:

“In September 2005, unauthorised clearance of mangroves took place in the Moanaanuanu Estuary, Whangamata Harbour. Two hectares of mangrove vegetation were removed on this occasion; a subset of which was also mown with a tractor to remove seedlings and pneumatophores (above-ground breathing roots).”

Furthermore, a previous study by Stokes et al. (2009) indicated that not only tree stems, but also the propagules established on the tidal flats were cut before they were incinerated. The study states:

“In recent years some local councils and community groups have been granted approval to remove fringe mangrove vegetation and any propagules that establish on the tidal flats. Only above-ground vegetation was cut (including pneumatophores), put into piles on-site, and later incinerated.”

These references support that our approach represents mangrove management in New Zealand removing both mangrove stems and aboveground root structures. However, we acknowledge that in other cases there may still be some remaining root structures, which could increase the hydraulic resistance. The duration for which these aboveground roots persist can vary. Research conducted on mangrove removal sites indicated that these fresh roots would be decomposed in less than 2 years when present at the sediment surface (Gladstone-Gallagher et al. 2014, Stokes and Harris 2015). However, in other geological settings such as coastal mangroves without riverine flow, the decomposition process could take longer (Ouyang et al. 2021, Stokes et al. 2023).

Considering these factors, we conducted five additional numerical experiments to investigate the impacts of remaining roots after mangrove removal for different lengths of time, compared to scenarios without any remaining roots, which is the approach adopted by this study (i.e. reference run). Specifically, we remove the vegetation stems at the beginning of the simulation (i.e. year 501) but allowed the roots to remain on the ground for an additional 2, 5, 15, 20 and 50 years such that each

scenario contains two periods differentiated by the presence of roots (Table R1). We also evaluate the infilling of accommodation space and changes of muddy regions for these two specific periods relative to the reference run. A relative value lower than 1 represents a slower infilling process, while a relative value larger than 1 refers to a faster infilling rate (bars in Fig. R6).

Table R1 (Table S3). Modelling scenarios testing the impacts of root persistence

Run ID	Model period I (no stems, with roots)	Model period II (no stems, no roots)	Legend in the Fig. R6
1	-	501-600	year 500 (reference)
2	501-502	503-600	year 502
3	501-505	506-600	year 505
4	501-515	516-600	year 515
5	501-520	521-600	year 520
6	501-550	551-600	year 550

As the reviewer suggested, our model experiments indicate that the ongoing presence of mangrove roots after mangrove removal would limit the extent of estuarine infilling (indicated by the green bars in Fig. R6). At the same time, some interesting behaviours emerge depending on the duration of root persistence. Estuarine infilling and muddy area development after short root persistence periods of 2 or 5 years follow similar trends as the reference scenario (as can be expected). However, when roots persist for longer (20 or 50 years), the rates of estuarine infilling and muddy area formation are slowed down initially but eventually exceed that of the reference run. These variations in the development of the estuarine morphology within the model suggest that when roots disappear after a multi-decadal delay, mud accumulation in fact accelerates.

To clarify the potential effects of root persistence, we have included detailed information and references to the corresponding figure in the method section.

Lines 624-637:

Impacts of different management approaches were then tested (Fig. 8), including mangrove removal according to different coverage reductions (i.e. 25%, 50% and 100% mangrove removal). *Mangrove removal was conducted by completely removing both stems and roots, following local studies documenting mangrove removal practices in New Zealand (Stokes et al. 2009, Swales et al. 2009). A sensitivity analysis was carried out to evaluate the impacts of root persistence on estuarine infilling processes as shown in Fig. S12. Leaving roots in place for longer periods of time initially constrains estuarine infilling, however, the subsequent disappearance of roots (due to decomposition) then accelerates estuarine infilling and the formation of muddy areas. Under short root persistence periods such as 2 or 5 years based on field observation (Gladstone-Gallagher et al. 2014, Stokes and Harris 2015), estuarine infilling and muddy area development follow similar trends as the reference in which both stems and roots were removed completely. Thus, the influence of root persistence on general trends in estuarine dynamics is limited.* We also conducted scenarios in which mud supply was reduced to pre-disturbance and intermediate levels, after different disturbance durations.

Figure R6. (Figure S12) Comparisons of changes in accommodation space (a) and muddy region (b) under different root persistence scenarios. The reference scenario applied in this comparison is based on Fig. 2k where both stems and roots are removed in year 500. As a comparison, different root persistence periods after mangrove removal, such as 2, 5, 15, 20 and 50 years, are shown as solid lines with different colors and has been indicated in Table R1. The inset plot in both panel (a) and (b) represents the relative (abbreviated as rel.) rate regarding the changes in accommodation space or muddy region fraction in two model periods differentiated by presence of roots. Specifically, a rate smaller than 1 means the examined scenario has a smaller infilling rate than the reference scenario in that period.

3.3

Also, one probable short-term impact of mangrove removal is likely to be an increase in accommodation, due to below-ground root collapse (See Cahoon et al. 2003; Lang'at et al. 2014).

R3.3

We agree that the decomposition of below-ground root materials following mangrove dieback can lead to a loss of surface elevation, thereby increasing accommodation space. It has been observed that tropical mangrove sites experience rapid and significant surface subsidence within one year of mangrove death (Cahoon et al. 2003, Lang'at et al. 2014).

However, in the context of mangrove removal sites in New Zealand, immediate sediment organic content reduction or elevation loss has not been observed thus far (Bulmer et al. 2017, Horstman et al. 2018, Stokes et al. 2023). There are two factors that can help explain these observations. Firstly, the decomposition rate of organic materials belowground may vary depending on vegetation species and local environmental conditions, such as tidal inundation and latitude of the mangrove system (Gladstone-Gallagher et al. 2014, Bulmer et al. 2017). In New Zealand, where the dominant mangrove species is *A. marina*, limited organic content reduction was observed at 36 months after mangrove removal, while a decay of 33% within the same period has been observed for tropical *Rhizophora*

mangrove roots (Siple and Donahue 2013). One study led by Lundquist et al. (2014) examined root biomass across nearly 40 mangrove removal sites in New Zealand, indicating that belowground biomass may persist as long as 16 years after mangrove removal. These findings align with an *in situ* decomposition experiment that demonstrated decadal delays in decomposition of belowground biomass, coupled with low rates of sediment erosion and surface elevation change at most mangrove removal sites in New Zealand estuaries (Gladstone-Gallagher et al. 2014).

Secondly, the belowground root biomass is generally lower in New Zealand than in other mangrove sites. According to the observations at the Mangawhai Harbour Estuary, northern New Zealand, the belowground root biomass was around 11 to 14 kg/m², which is much lower than for example mangrove sites in Australia, where belowground root biomass has been reported to range from 10 to 40 kg/m² (Tran et al. 2017). Previous studies highlight that in highly organic soils, root growth and decay and organic matter accumulation can significantly influence surface elevation (McKee 2011, Krauss et al. 2014). However, in the case of mangrove removal sites in New Zealand, organic contents and root production is limited, ~ 50 g/m²/yr, contributing less than 1% of the sedimentation volume (Swales et al. 2019). It is important to note that this root production rate is much lower than that observed in other sites, which will be introduced in section R3.4. This implies that root collapse would only have a small effect on surface elevation change in New Zealand.

In conclusion, we believe that the significance of the belowground root collapse in driving surface elevation changes is contingent upon the specific geographic location. In New Zealand mangrove forests, given the slow decomposition rate and limited belowground root biomass, the contribution of surface elevation loss due to root collapse is likely to be negligible. Consequently, we consider this process to be of less importance in our analysis. To address this important aspect, we have explicitly discussed the potential effects of belowground root collapse:

Lines 384-390:

Mangrove dieback has also been found to drive root collapse, which in turn lowers surface elevation and increases accommodation space (Cahoon et al. 2003, Lang'at et al. 2014). However, studies in New Zealand indicate that limited changes occurred in the surface elevation after mangrove removal (Bulmer et al. 2017, Horstman et al. 2018, Stokes et al. 2023). This is probably because of slow decomposition rates in soil organic matter (Gladstone-Gallagher et al. 2014, Lundquist et al. 2014) and limited belowground root biomass as well as root production compared to other tropical mangrove sites (Tran et al. 2017, Swales et al. 2019).

3.4

(iii) Mangroves reduce accommodation through vertical root growth, which can be the dominant contributor to elevation gain (see McKee et al. 2007 in a different context, but making this point).

R3.4

We concur that root production can be crucial (McKee et al. 2007), and at some sites it can be the dominant process contributing to surface accretion (McKee 2011, Lovelock et al. 2015). Nevertheless, as we pointed out in R3.3, measurements conducted at the mangrove site in the Firth of Thames, New Zealand, indicate a relatively low root production, with the maximum rate of approximately 50 g/m²/yr, contributing less than 1% sediment volume (Swales et al. 2019). This magnitude of root production is considerably lower than what has been observed in other mangrove sites outside New Zealand. For

instance, McKee et al. (2007) measured root production rates ranging from 80 to 600 g/m²/yr in the *R. mangle* forests at Twin Cays, Belize. The observed disparity in root production between New Zealand mangrove forests and other tropical mangrove sites could potentially be attributed to variations in mangrove species and/or latitudes.

To understand the potential impacts of root production on surface accretion, we compiled published data and developed correlations for each data sample, as presented in Fig. R7. These published data indicate the accumulation of roots (live and dead) based on the same technique, i.e. ingrowth bags (McKee et al. 2007, McKee 2011, Lovelock et al. 2015, Swales et al. 2019). The analysis reveals a linear relationship between root production and surface accretion, whereby greater root production leads to a faster rate of accretion. Also, low levels of root production, such as 50 g/m²/yr or less, result in a negligible accretion rate, which can be significantly lower than 0.5 mm/yr. By comparing this minimal belowground root-driven surface accretion rate with the mineral sediment accretion rate (Fig. 4e-f in the main text), which is estimated to be 10 to 40 times greater, we conclude that the contribution of root growth to elevation gain in the New Zealand mangrove forest site is negligible.

To ensure clarity for the reader, we have included the compiled data (Table R2) and the new plot (Fig. R7) in the discussion section to elucidate this point further.

Lines 376-384:

In this study, the infilling of accommodation space is fully driven by mineral sedimentation, while organic accretion driven by root production is not included. We acknowledge that belowground root growth can be an important or even dominant process controlling surface elevation change (McKee et al. 2007). According to field observations from multiple mangrove sites, belowground root induced surface accretion tends to show a linear relationship with root production (Fig. S13). However, local data shows that belowground root production in New Zealand mangrove forests is smaller (50 g/m²/yr) than in most other tropical mangrove sites, contributing less than 1% sedimentation volume (Swales et al. 2019). Such a limited root production is expected to result in a negligible accretion rate (less than 0.5 mm/yr, Fig. S13).

Figure R7. (Figure S13) Relations between the root production and surface accretion rates. Sample points are extracted from previous field observations summarized in Table R2. Linear regression lines are applied to each data set.

Table R2. (Table S4) Summary of field data on root accumulation and accretion rate

No.	Root production (g/m ² /yr)	Accretion rate (mm/yr)	Site	Reference
1	649.36	8.77	Twin Cays, Belize and Rookery Bay, Florida	McKee (2011)
2	269.26	4.28		
3	80.52	1.24		
4	648.05	8.49		
5	111.97	1.54		
6	153.92	2.4		
7	794.85	11.81		
8	130.2	3.3	Moreton Bay, Queensland	Lovelock et al. (2015)
9	604.8	9.3	Fine roots at Twin Cays, Belize	McKee et al. (2007)
10	37	0.71		
11	43	0.832		
12	189	3.753		
13	164	3.266		
14	260	5.172		
15	168	3.306		
16	197	3.915		

17	310	6.187	Coarse roots at Twin Cays, Belize	McKee et al. (2007)
18	339	6.755		
19	45	0.446		
20	39	0.324		
21	205	1.987		
22	316	3.123		
23	163	1.583		
24	132	1.339		
25	328	3.286		
26	194	1.906		
27	403	3.975		

3.5

One final point is that estuarine environments across the globe are subject to increasing accommodation due to accelerating sea-level rise. Over coming decades this may become a counter-balancing effect on catchment sediment delivery (which is surely a pulse event). In many locations mangroves are expanding due to increased accommodation (moving landward into saltmarshes: Kelleway et al 2017), rather than decreased accommodation. This broader context could be considered in the discussion.

R3.5

Many thanks for this suggestion. In this study we did not incorporate the influence of sea-level rise since the current local sea-level rise rates are relatively low (approximately 1.4 mm/yr), compared to the high sedimentation rate of 10-100 mm/yr (Denys et al. 2020). However, considering the projected accelerated sea-level rise in the future, particularly in estuaries with lower sedimentation rates, it is plausible that the sea-level rise rate could surpass the local sedimentation rate, leading to an increase in both vertical and horizontal accommodation space (Schuerch et al. 2018, Rogers 2021). Thus, as sea levels rises, the rapid infilling of accommodation space in New Zealand estuaries could indeed gradually decelerate. We have now included a discussion on the impacts of sea-level rise in our revised manuscript.

Lines 390-398:

Apart from belowground processes, sea-level rise can also affect changes in accommodation space. Here we did not consider the impacts of sea-level rise on estuarine infilling given the low sea-level rise rate (around 1.4 mm/yr), compared to the high sedimentation rate (10-100 mm/yr) (Denys et al. 2020). However, projected accelerations in sea-level rise may create additional accommodation space (Schuerch et al. 2018, Rogers 2021), thus slowing down the estuarine infilling process. Sedimentation rates along vegetated tidal flats have been found to be non-linearly related to sea-level rise rates (Xie et al. 2022), such that future estuarine infilling is likely to be a complex process that needs to be further explored (Boechat Albernaz et al. 2023).

Minor points

3.6

Fig 1. Make clear that mangrove coverage (%) refers to percentage of estuarine area (a reader may assume you mean percentage increase against original extent)

R3.6

In our paper, mangrove coverage indeed refers to the percentage of estuarine area. We now added an extra sentence in the caption of Figure 1 to illustrate the meaning of mangrove coverage.

Lines 158-159:

... riverine input. *Mangrove coverage refers to the percentage of mangrove presence relative to the estuarine area.* Mangrove coverage data

3.7

Fig 3. Could the x107 relating to the units of the y-axis appear in the axis title? I missed it the first time

R3.7

Thanks for spotting this. See the new Fig. 3 below, we have set the y-axis as Accommodation space ($m^3; \times 10^7$).

3.8

Line 120. Consider citing the pioneering work of Bruce Thom here- your point about mangroves being a more passive element in estuarine geomorphology was emphatically made in his early work (e.g. Thom 1967).

R3.8

Thanks, we have now incorporated a reference to the work by Thom (1967).

Lines 126-128:

Furthermore, rather than significantly increasing sedimentation rates, it has been suggested that mangroves are opportunistic and colonize areas that have already reached a suitable intertidal elevation through historic sedimentation (*Thom, 1967*; Swales et al., 2015).

3.9

Line 320. I couldn't find Wu et al. 2001 or Montgomery et al. 2022 in the reference list. These are very important references in the context of your argument, and I would have liked to refer to them.

R3.9

We thank the reviewer for pointing out these missing references. We now add these two missing references to the reference list.

Lines 812-813 and 906-907:

Wu, Y., Falconer, R. A., & Struve, J. (2001). Mathematical modelling of tidal currents in mangrove forests. *Environmental Modelling & Software*, 16(1), 19-29.

Montgomery, J., Bryan, K., R. & Coco, G. (2022). The role of mangroves in coastal flood protection: The importance of channelization. *Continental Shelf Research*, 243, 104762.

3.10

Figure 6. You refer to “positive feedbacks” in this diagram when I think you mean relationships. For example, the relationship between soil erosion and sediment delivery to the coast is not a “positive feedback”, because sediment delivery to the coast has no influence on soil erosion.

R3.10

We agree with the reviewer that the currently so-called ‘positive feedback’ is a bit misleading because it is actually a one-way link between two processes, such as ‘soil erosion’ and ‘sediment delivery to the coast’. To avoid any confusion, we now use ‘positive link and ‘negative link to describe the linkages within the diagram following the terminology used in previous studies on the evolution of estuaries and tidal embayments (de Haas et al. 2018), see the new Figure 6. We also checked the text and find that these changes do not influence our meaning.

In addition, to make clear the meaning of ‘positive and negative links’, we now include more details in the caption of Figure 6. See the new Figure 6 and the corresponding caption below.

Fig. 6 Conceptual diagram outlining distinct bio-morphodynamic and anthro-bio-morphodynamic feedbacks at the local, estuary and source-to-sink scale. Reinforcing and balancing feedback loops (Payo et al., 2016) are indicated. Here, ‘upstream land-use’ refers to pastoral

farming and agriculture which result in large-scale catchment deforestation. *'Positive link' and 'negative link' refer to positive and negative correlations, respectively.* Dashed lines surrounding the rectangles for 'Human interventions' represent either a one-time intervention or more continuous interventions that trigger the feedback loop. The dashed line indicating the ~~feedback between~~ *link from* 'sediment delivery to the coast' ~~and to~~ 'upstream land-use' suggests a more sustainable and effective management approach that addresses source-to-sink linkages.

References

- Cahoon, D. R., Hensel, P., Rybczyk, J., McKee, K. L., Proffitt, C. E., & Perez, B. C. (2003). Mass tree mortality leads to mangrove peat collapse at Bay Islands, Honduras after Hurricane Mitch. *Journal of ecology*, 91(6), 1093-1105.
- Kelleway, J. J., Cavanaugh, K., Rogers, K., Feller, I. C., Ens, E., Doughty, C., & Saintilan, N. (2017). Review of the ecosystem service implications of mangrove encroachment into salt marshes. *Global Change Biology*, 23(10), 3967-3983.
- Lang'at, J. K. S., Kairo, J. G., Mencuccini, M., Bouillon, S., Skov, M. W., Waldron, S., & Huxham, M. (2014). Rapid losses of surface elevation following tree girdling and cutting in tropical mangroves. *Plos one*, 9(9), e107868.
- McKee, K. L., Cahoon, D. R., & Feller, I. C. (2007). Caribbean mangroves adjust to rising sea level through biotic controls on change in soil elevation. *Global Ecology and Biogeography*, 16(5), 545-556.
- Thom, B. G. (1967). Mangrove ecology and deltaic geomorphology: Tabasco, Mexico. *The Journal of Ecology*, 301-343.
- Woodroffe, C. D., Rogers, K., McKee, K. L., Lovelock, C. E., Mendelssohn, I. A., & Saintilan, N. (2016). Mangrove sedimentation and response to relative sea-level rise. *Annual review of marine science*, 8, 243-266

We thank the reviewer for listing these references.

References used in this response letter.

- Boechat Albernaz, M., M. Z. M. Brückner, B. van Maanen, A. J. F. van der Spek and M. G. Kleinmans (2023). "Vegetation Reconfigures Barrier Coasts and Affects Tidal Basin Infilling Under Sea Level Rise." *Journal of Geophysical Research: Earth Surface* **128**(4): e2022JF006703.
- Boechat Albernaz, M., L. Roelofs, H. J. Pierik and M. G. Kleinmans (2020). "Natural levee evolution in vegetated fluvial-tidal environments." *Earth Surface Processes and Landforms* **45**(15): 3824-3841.
- Bulmer, R. H., M. Lewis, E. O'Donnell and C. J. Lundquist (2017). "Assessing mangrove clearance methods to minimise adverse impacts and maximise the potential to achieve restoration objectives." *New Zealand Journal of Marine and Freshwater Research* **51**(1): 110-126.
- Cahoon, D. R., P. Hensel, J. Rybczyk, K. L. McKee, E. Proffitt and P. B. C. (2003). "Mass tree mortality leads to mangrove peat collapse at Bay Islands, Honduras after Hurricane Mitch." *Journal of Ecology* **91**: 1093-1105.
- de Haas, T., H. Pierik, A. Van der Spek, K. Cohen, B. Van Maanen and M. Kleinmans (2018). "Holocene evolution of tidal systems in The Netherlands: Effects of rivers, coastal boundary conditions, eco-engineering species, inherited relief and human interference." *Earth-Science Reviews* **177**: 139-163.
- Denys, P. H., R. J. Beavan, J. Hannah, C. F. Pearson, N. Palmer, M. Denham and S. Hreinsdottir (2020). "Sea Level Rise in New Zealand: The Effect of Vertical Land Motion on Century-Long Tide Gauge Records in a Tectonically Active Region." *Journal of Geophysical Research: Solid Earth* **125**(1): e2019JB018055.
- Fagherazzi, S., M. L. Kirwan, S. M. Mudd, G. R. Guntenspergen, S. Temmerman, A. D'Alpaos, J. van de Koppel, J. M. Rybczyk, E. Reyes, C. Craft and J. Clough (2012). "Numerical models of salt marsh evolution: Ecological, geomorphic, and climatic factors." *Reviews of Geophysics* **50**(1): RG1002.
- Gladstone-Gallagher, R. V., C. J. Lundquist and C. A. Pilditch (2014). "Mangrove (*Avicennia marina* subsp. *australasica*) litter production and decomposition in a temperate estuary." *New Zealand Journal of Marine and Freshwater Research* **48**(1): 24-37.
- Horstman, E. M., C. J. Lundquist, K. R. Bryan, R. H. Bulmer, J. C. Mullarney and D. J. Stokes (2018). The dynamics of expanding mangroves in New Zealand. *Threats to Mangrove Forests: Hazards, Vulnerability, and Management*. C. Makowski and C. W. Finkl, Springer: 23-52.
- Kirwan, M. L. and J. P. Megonigal (2013). "Tidal wetland stability in the face of human impacts and sea-level rise." *Nature* **504**(7478): 53-60.
- Kirwan, M. L., A. B. Murray, J. P. Donnelly and D. R. Corbett (2011). "Rapid wetland expansion during European settlement and its implication for marsh survival under modern sediment delivery rates." *Geology* **39**(5): 507-510.
- Krauss, K. W., K. L. McKee, C. E. Lovelock, D. R. Cahoon, N. Saintilan, R. Reef and L. Chen (2014). "How mangrove forests adjust to rising sea level." *New Phytol* **202**(1): 19-34.
- Lang'at, J. K. S., J. G. Kairo, M. Mencuccini, S. Bouillon, M. W. Skov, S. Waldron and M. Huxham (2014). "Rapid losses of surface elevation following tree girdling and cutting in tropical mangroves." *Plos one* **9**(9): e107868.
- Lovelock, C. E., M. F. Adame, V. Bennion, M. Hayes, R. Reef, N. Santini and D. R. Cahoon (2015). "Sea level and turbidity controls on mangrove soil surface elevation change." *Estuarine, Coastal and Shelf Science* **153**: 1-9.
- Lovelock, C. E., D. R. Cahoon, D. A. Friess, G. R. Guntenspergen, K. W. Krauss, R. Reef, K. Rogers, M. L. Saunders, F. Sidik, A. Swales, N. Saintilan, L. X. Thuyen and T. Triet (2015). "The vulnerability of Indo-Pacific mangrove forests to sea-level rise." *Nature* **526**(7574): 559-563.
- Lundquist, C., D. Morrisey, R. Gladstone-Gallagher and A. Swales (2014). Managing Mangrove Habitat Expansion in New Zealand. *Mangrove Ecosystems of Asia*. I. Faridah-Hanum, A. Latiff, K. R. Hakeem and M. Ozturk. New York, Springer: 415-438.
- McKee, K. L. (2011). "Biophysical controls on accretion and elevation change in Caribbean mangrove ecosystems." *Estuarine, Coastal and Shelf Science* **91**(4): 475-483.
- McKee, K. L., D. R. Cahoon and I. C. Feller (2007). "Caribbean mangroves adjust to rising sea level through biotic controls on change in soil elevation." *Global Ecology and Biogeography* **16**(5): 545-556.

- Nienhuis, J. H., A. D. Ashton, D. A. Edmonds, A. J. F. Hoitink, A. J. Kettner, J. C. Rowland and T. E. Törnqvist (2020). "Global-scale human impact on delta morphology has led to net land area gain." *Nature* **577**(7791): 514-518.
- Ouyang, X., F. Guo and S. Y. Lee (2021). "The impact of super-typhoon Mangkhut on sediment nutrient density and fluxes in a mangrove forest in Hong Kong." *Science of The Total Environment* **766**: 142637.
- Rogers, K. (2021). "Accommodation space as a framework for assessing the response of mangroves to relative sea-level rise." *Singapore Journal of Tropical Geography* **42**(2): 163-183.
- Schepers, L., A. Van Braeckel, T. J. Bouma and S. Temmerman (2020). "How progressive vegetation die-off in a tidal marsh would affect flow and sedimentation patterns: A field demonstration." *Limnology and Oceanography* **65**(2): 401-412.
- Schuerch, M., T. Spencer, S. Temmerman, M. L. Kirwan, C. Wolff, D. Lincke, C. J. McOwen, M. D. Pickering, R. Reef, A. T. Vafeidis, J. Hinkel, R. J. Nicholls and S. Brown (2018). "Future response of global coastal wetlands to sea-level rise." *Nature* **561**(7722): 231-234.
- Siple, M. C. and M. J. Donahue (2013). "Invasive mangrove removal and recovery: Food web effects across a chronosequence." *Journal of Experimental Marine Biology and Ecology* **448**: 128-135.
- Stokes, D., T. Healy and P. Cooke (2009). "Surface elevation changes and sediment characteristics of intertidal surfaces undergoing mangrove expansion and mangrove removal, Waikaraka Estuary, Tauranga Harbour, New Zealand." *International Journal of Ecology and Development* **12**.
- Stokes, D. J., H. E. Glover, K. R. Bryan and C. A. Pilditch (2023). "Environmental state of a small intertidal estuary a decade after mangrove clearance, Waikaraka Estuary, Aotearoa New Zealand." *Ocean & Coastal Management* **243**: 106731.
- Stokes, D. J. and R. J. Harris (2015). "Sediment properties and surface erodibility following a large-scale mangrove (*Avicennia marina*) removal." *Continental Shelf Research* **107**: 1-10.
- Swales, A., R. Bell, R. Gorman, J. Oldman, A. Altenberger, C. Hart, L. Claydon, S. Wadhwa and R. Ovenden (2009). "Potential future changes in mangrove-habitat in Auckland's east-coast estuaries." *NIWA Client Report: HAM2008-030 prepared for Auckland Regional Council*.
- Swales, A., G. Reeve, D. R. Cahoon and C. E. Lovelock (2019). "Landscape Evolution of a Fluvial Sediment-Rich *Avicennia marina* Mangrove Forest: Insights from Seasonal and Inter-annual Surface-Elevation Dynamics." *Ecosystems* **22**(6): 1232-1255.
- Temmerman, S., G. Govers, P. Meire and S. Wartel (2004). "Simulating the long-term development of levee-basin topography on tidal marshes." *Geomorphology* **63**(1-2): 39-55.
- Thom, B. G. (1967). "Mangrove ecology and deltaic Geomorphology: Tabasco, Mexico." *British Ecology Society* **55**(2): 301-343.
- Tran, P., I. Gritcan, J. Cusens, A. C. Alfaro and S. Leuzinger (2017). "Biomass and nutrient composition of temperate mangroves (*Avicennia marina* var. *australasica*) in New Zealand." *New Zealand Journal of Marine and Freshwater Research* **51**(3): 427-442.
- van Maanen, B., G. Coco and K. R. Bryan (2015). "On the ecogeomorphological feedbacks that control tidal channel network evolution in a sandy mangrove setting." *Proc Math Phys Eng Sci* **471**(2180): 20150115.
- Woodroffe, C. D., K. Rogers, K. L. McKee, C. E. Lovelock, I. A. Mendelssohn and N. Saintilan (2016). "Mangrove Sedimentation and Response to Relative Sea-Level Rise." *Ann Rev Mar Sci* **8**: 243-266.
- Xie, D., C. Schwarz, M. G. Kleinhans, Z. Zhou and B. van Maanen (2022). "Implications of Coastal Conditions and Sea-Level Rise on Mangrove Vulnerability: A Bio-Morphodynamic Modeling Study." *Journal of Geophysical Research: Earth Surface* **127**(3): e2021JF006301.

Reviewer #3 (Remarks to the Author):

I would like to thank the authors for doing such a thorough job of responding to my earlier comments, including additional analyses. These detailed and thoughtful responses have resolved the concerns raised. I recommend publication.

Dear reviewers,

We would like to express our gratitude to all three reviewers for their excellent feedback on our manuscript. The manuscript has undergone significant improvement during the revision process, and we are thankful for the engagement and critical insights provided by the reviewers.

Sincerely,
Danghan Xie, on behalf of all the authors,

REVIEWER COMMENTS

Reviewer #3 (Remarks to the Author):

I would like to thank the authors for doing such a thorough job of responding to my earlier comments, including additional analyses. These detailed and thoughtful responses have resolved the concerns raised. I recommend publication.